# N-terminal tyrosine of ISCU2 triggers [2Fe-2S] cluster synthesis by ISCU2 dimerization

Sven-A. Freibert [1,2,6], Michal T. Boniecki[3,6], Claudia Stümpfig[1], Vinzent Schulz[1], Nils Krapoth[1], Dennis R. Winge[1,4], Ulrich Mühlenhoff[1], Oliver Stehling[1,2], Miroslaw Cygler [3✉] & Roland Lill [1,2,5✉]

Synthesis of iron-sulfur (Fe/S) clusters in living cells requires scaffold proteins for both facile synthesis and subsequent transfer of clusters to target apoproteins. The human mitochondrial ISCU2 scaffold protein is part of the core ISC (iron-sulfur cluster assembly) complex that synthesizes a bridging [2Fe-2S] cluster on dimeric ISCU2. Initial iron and sulfur loading onto monomeric ISCU2 have been elucidated biochemically, yet subsequent [2Fe-2S] cluster formation and dimerization of ISCU2 is mechanistically ill-defined. Our structural, bio-chemical and cell biological experiments now identify a crucial function of the universally conserved N-terminal Tyr35 of ISCU2 for these late reactions. Mixing two, per se non-functional ISCU2 mutant proteins with oppositely charged Asp35 and Lys35 residues, both bound to different cysteine desulfurase complexes NFS1-ISD11-ACP, restores wild-type ISCU2 maturation demonstrating that ionic forces can replace native Tyr-Tyr interactions during dimerization-induced [2Fe-2S] cluster formation. Our studies define the essential mechanistic role of Tyr35 in the reaction cycle of de novo mitochondrial [2Fe-2S] cluster synthesis.

[1] Institut für Zytobiologie im Zentrum SYNMIKRO, Philipps-Universität Marburg, Karl-von-Frisch-Str. 14, 35032 Marburg, Germany. [2] Core Facility 'Protein Biochemistry and Spectroscopy', Karl-von-Frisch-Str. 14, 35032 Marburg, Germany. [3] Department of Biochemistry, Microbiology & Immunology, University of Saskatchewan, 107 Wiggins Rd, Saskatoon, SK S7N 5E5, Canada. [4] Department of Medicine, University of Utah Health Sciences Center, Salt Lake City, UT, USA. [5] LOEWE Zentrum für Synthetische Mikrobiologie SynMikro, Hans-Meerwein-Str., 35043 Marburg, Germany. [6] These authors contributed equally: Sven-A. Freibert, Michal T. Boniecki. ✉email: miroslaw.cygler@usask.ca; lill@staff.uni-marburg.de

I ron-sulfur (Fe/S) clusters are ancient metallo-cofactors that execute versatile physiological functions in, e.g., antiviral defense, iron regulation, respiration, citric acid cycle, and numerous metabolic reactions[1–4]. Moreover, basic cellular processes such as protein translation or DNA synthesis and repair depend on numerous, usually essential Fe/S proteins[5–7]. Given this broad involvement of Fe/S clusters in cellular physiology, it is not surprising that their in vivo synthesis and insertion into apoproteins is a highly conserved process, and usually essential for cell viability[8–10]. In (non-green) eukaryotes, cellular Fe/S protein biogenesis is executed by the mitochondrial iron-sulfur cluster assembly (ISC) machinery with up to 18 constituents, and the cytosolic iron-sulfur protein assembly (CIA) machinery with 13 known proteins[4,11,12]. These two machineries are connected via the mitochondrial ABC transporter ABCB7 which exports a yet unknown, sulfur- and possibly iron-containing component for usage by the CIA system[13–16]. Genetic mutations in the biogenesis proteins cause numerous severe human diseases[17–19].

The biogenesis process catalyzed by both the ISC and CIA machineries can formally be dissected into three distinct steps; de novo Fe/S cluster synthesis, trafficking via transfer proteins, and insertion into recipient apoproteins via targeting factors[10]. The initial synthesis reaction always occurs on dedicated scaffold proteins that provide a favorable biochemical platform for both rapid cluster assembly and facile dislocation for trafficking to downstream biogenesis components[20,21]. Prominent representatives of Fe/S scaffolds are yeast Isu1 and human ISCU2, mitochondria-localized members of the highly conserved IscU protein family that originated from bacteria[8,22,23]. IscU-like proteins possess three conserved Cys residues that are all essential for de novo generation of a bridging [2Fe-2S] cluster on IscU or ISCU2 dimers[24,25]. The synthesis of [4Fe-4S] clusters occurs later in the mitochondrial ISC pathway, involving the ISCA1-ISCA2-IBA57 complex for reductive fusion of ISCU2-derived [2Fe-2S] clusters[26,27]. Numerous biochemical and structural studies on both bacterial and mitochondrial IscU-like scaffold proteins have provided mechanistic insights into [2Fe-2S] cluster synthesis and the function of the involved ISC components[4]. Together, they form the 'core ISC complex' whose 3D architecture has been partially resolved by X-ray crystallography, electron cryo-microscopy (cryo-EM) and small angle X-ray scattering[28–30]. The central component of this multimeric core ISC complex, the dimeric pyridoxal phosphate (PLP)-dependent cysteine desulfurase NFS1 with its two bound regulatory proteins ISD11 and ACP, initially generates an enzyme-bound persulfide (-SSH) from free cysteine[31,32]. The ISC protein frataxin (FXN) allosterically activates persulfide transfer from NFS1 to one of the three conserved Cys residues of iron-containing ISCU2, binding to opposite tips of the NFS1 dimer[33–36]. Persulfide reduction by the ferredoxin FDX2 (and its reductase FDXR) generates the sulfide required for [2Fe-2S] cluster formation[25,36,37]. While these initial steps have been resolved to some extent, the molecular mechanisms underlying the actual [2Fe-2S] cluster synthesis are largely unknown, even though the end product, i.e., dimeric ISCU2 holding a bridging [2Fe-2S] cluster, has been defined by in vitro enzymatic reconstitution experiments[24,25,38]. Open questions concerning [2Fe-2S] cluster synthesis include the problems of how the second set of iron and sulfide ions are acquired to form the [2Fe-2S] cluster, how ISCU2 dissociates from NFS1, and how ISCU2 dimers are formed. A possible intermediate of these terminal reactions may be represented by a crystal structure of a NFS1-ISCU2-related archaeal IscS-IscU complex which binds a shared [2Fe-2S] cluster[39]. However, it has been noted that IscS within this complex is not a functional desulfurase[40], raising questions about the sulfur source and the physiological relevance of this putative intermediate.

The work reported here is concentrated on the mechanistic role of the N-terminal Tyr35 residue of human ISCU2 that is universally conserved in all mitochondrial and bacterial IscU-like proteins[20,41,42]. The physiological importance of this residue has been shown in vivo but the molecular reason for its essential function is unknown. To help clarify its function, we resolved several new crystal structures of the (NIAU2)$_2$ complex (i.e., dimeric NFS1-ISD11-ACP-ISCU2), and performed biochemical and cell biological experiments with ISCU2 and ISCU2-Tyr35 mutant proteins. These studies revealed the fundamental mechanistic role of Tyr35 in the final steps of [2Fe-2S] cluster synthesis on an ISCU2 dimer. Tyr35-Tyr35 interactions between two iron- and sulfide-containing ISCU2 bound to the (NIA)$_2$ complex were essential for both ISCU2 dimerization and accompanying [2Fe-2S] cluster formation. This mechanism may represent a general principle of how IscU-like proteins utilize their N-terminal Tyr residue for dimerization-triggered de novo [2Fe-2S] cluster synthesis.

## Results

**Crucial in vivo function of N-terminal Tyr35 of ISCU2.** The physiological importance of the universally conserved Tyr35 of human ISCU2 (Supplementary Fig. 1a,b) was analyzed by an RNAi depletion-complementation approach in which first the ISCU2 mRNA was depleted by short hairpin RNA (shISCU) for 3 and 6 days by repeated transfection of HeLa cells[43,44] (Fig. 1a). The resulting mitochondrial ISCU2 deficiency expectedly led to a severe diminution in both protein levels and activities of several mitochondrial Fe/S proteins including aconitase, succinate dehydrogenase (SDH), and ferrochelatase, while heme-containing cytochrome oxidase (COX) was hardly affected[23] (Fig. 1b, c; Supplementary Table 1). Second, ISCU2-depleted HeLa cells were complemented with plasmids encoding silently mutated, RNAi-resistant versions of ISCU2-Tyr35Ala (smISCU2_Y35A) or ISCU2 containing (smISCU2) or lacking (smISCU2ΔMLS) the mitochondrial localization signal (MLS) as controls (Supplementary Fig. 1c). Digitonin fractionation and immunostaining revealed that both the mutant smISCU2_Y35A and wild-type smISCU2 were almost exclusively targeted to mitochondria and the MLS was cleaved (Fig. 1a). smISCU2ΔMLS was predominantly located in the cytosol fraction, yet a portion was found in the membrane fraction. This protein was sensitive to proteinase K digestion even without hypotonic swelling (a procedure leading to opening of the mitochondrial outer membrane) or detergent lysis (Supplementary Fig. 2). The protein therefore was associated with the mitochondrial surface rather than being sequestered into the matrix. Expression of smISCU2, smISCU2ΔMLS, or smISCU2_Y35A in ISCU2-depleted cells differentially complemented the observed strong defects in Fe/S protein levels and activities (Fig. 1b, c; Supplementary Table 1). While smISCU2 fully restored all Fe/S maturation defects, the smISCU2_Y35A mutant protein, and smISCU2ΔMLS as a control, did not significantly compensate any of these defects. In turn, this result confirmed that smISCU2ΔMLS was not localized to the matrix. These findings clearly demonstrate the crucial role of Tyr35 for mitochondrial ISCU2 function in cellular Fe/S protein biogenesis, nicely fitting to results for yeast Isu1-Isu2 and bacterial IscU[41,42]. Therefore, the universally conserved N-terminal Tyr in IscU-like proteins plays an important physiological role in Fe/S protein assembly in vivo, yet its exact mechanistic task remains elusive.

**High resolution crystal structures of NFS1-ISD11-ACP-ISCU2 complexes.** To gain structural insights into the location and potential role of Tyr35 within the core ISC complex, we

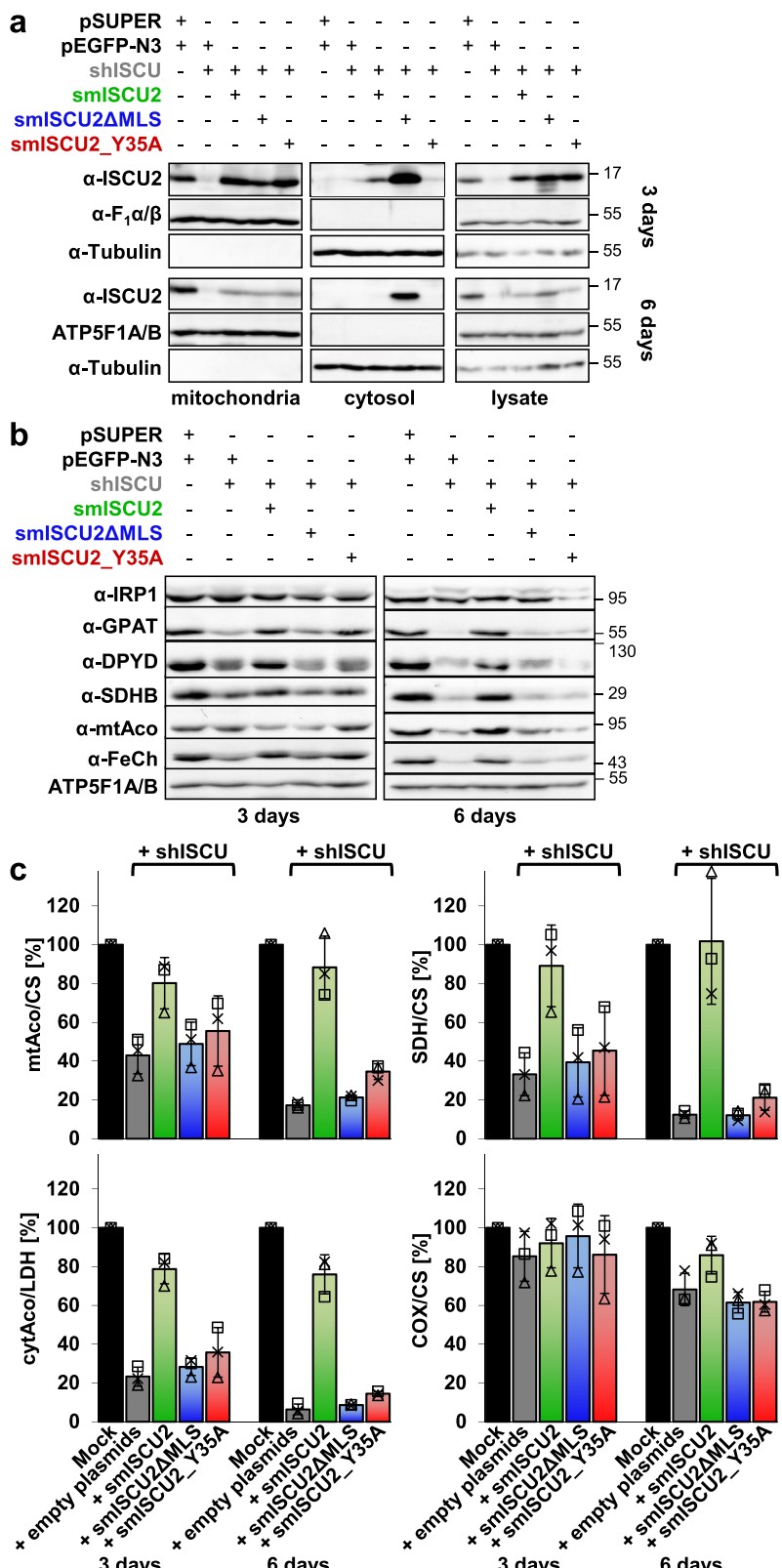

crystallized the dimeric (NIAU2)₂ complex containing wild-type or various mutant ISCU2 proteins (Supplementary Table 2). Since non-functional cytosolic ISCU1 lacks the conserved Tyr35[23], it was particularly interesting to compare the obtained structures to those of (NIAU1)₂ (i.e., dimeric NFS1-ISD11-ACP-ISCU1)[28]. The (NIAU2)₂ complexes were purified from *E. coli* lysates as previously reported for the (NIAU1)₂ complex[28], yet

crystallized under different conditions indicating a distinct behavior of the two complexes. The (NIAU2)₂ crystals showed higher order and, for wild-type ISCU2, diffracted up to 1.8 Å resolution, i.e., significantly better than the ~3.2 Å obtained for both the (NIAU1)₂ crystals and the cryo-EM structure of (NIAU2F)₂ with bound FXN[30] (PDB ID: 6NZU). The highest resolution (1.57 Å) was obtained for a complex containing the

**Fig. 1 The N-terminal Tyr35 of ISCU2 is essential for cellular Fe/S protein biogenesis.** HeLa cells were transfected twice with vector shISCU for depletion of ISCU2 or with control vectors (pSUPER, pEGFP-N3), as well as with plasmids encoding silently mutated (sm), RNAi-resistant versions of huISCU2 containing (smISCU2) or lacking (smISCU2ΔMLS) the mitochondrial localization sequence, or a silently mutated version of huISCU2_Y35A (smISCU2_Y35A). Cell growth was for 3 and 6 days. **a** Harvested cells were treated with digitonin, and the cell lysate was separated by centrifugation into cytosol and a membrane fraction containing mitochondria. Cellular ISCU2 protein expression was analyzed by immunoblotting. Signals for α and β subunits of mitochondrial $F_1$-ATP synthase ($F_1\alpha/\beta$) and cytosolic tubulin revealed the fractionation efficiency and served as loading controls. **b** Cell lysates were examined by immunostaining for the indicated Fe/S and reference proteins. Parts a,b show representative blots. For quantitative evaluation see Supplementary Table 1. **c** Membrane or cytosol fractions (depending on the protein analyzed) were analyzed for the indicated mitochondrial or cytosolic enzyme activities relative to citrate synthase (CS) or lactate dehydrogenase (LDH) activity, respectively (mean ± SD; $n = 3$ biological replicates). mtAco and cytAco, mitochondrial and cytosolic aconitase, COX cytochrome $c$ oxidase, SDH succinate dehydrogenase. Source Data are provided as a Source Data file.

ISCU2-M140I variant (Supplementary Table 2), which in yeast escapes the need for FXN[45]. The M140I mutation had no discernable effect on the global or local structure of ISCU2 or the entire complex, and thus did not provide any structural clues for the functional effect of this mutation in yeast. The $(NIAU2)_2$ crystal structures were solved by molecular replacement and found to be overall quite similar to both the $(NIAU1)_2$ X-ray and $(NIAU2F)_2$ cryo-EM structures, with some minor differences (Fig. 2a; Supplementary Fig. 3a). Nevertheless, the sidechain conformations were much better defined in the new $(NIAU2)_2$ X-ray structures. The backbone conformations of NFS1, ISD11, and ACP within the $(NIAU1)_2$ and $(NIAU2)_2$ structures (containing wild-type ISCU2) were nearly identical with a root-mean-square deviation (rmsd) for the Cα atoms of 0.34 Å for NFS1, 0.40 Å for ISD11, and 0.64 Å for ACP. Main differences for NFS1 were found in the partially disordered active Cys381-containing loop, consistent with its flexibility (Supplementary Fig. 3b). For ISD11-ACP, two more N- and one more C-terminal residues and two additional salt bridges were resolved in the high-resolution $(NIAU2)_2$ structure (Supplementary Fig. 3c–e).

Some critical differences were recorded for the two ISCU proteins. While residues 40-157 (ISCU2 numbering) of wild-type ISCU2 and ISCU1 superimposed well with an rmsd of 0.73 Å (Cα atoms) and 1.65 Å (all atoms) (Supplementary Fig. 3f), a ~6° rotation of the two ISCU proteins along the twofold axis relative to the NFS1 dimer was observed locating ISCU2 closer to the second NFS1' monomer (Fig. 2b). Comparison of crystal contacts in the two structures suggested that this difference may be the result of packing differences. A difference in orientation of similar order of magnitude is observed also in the cryo-EM structure of the $(NIAU2F)_2$ complex. More importantly, the ISCU1/2 structures differed significantly at their N termini, which were resolved from the first residue of mature ISCU2 (Tyr35) and from Leu10[ISCU1] (corresponding to Tyr35[ISCU2]; Supplementary Fig. 1), leaving the N-terminal nine residues of ISCU1 disordered (Supplementary Fig. 3f). Most interestingly, the electron density for Tyr35[ISCU2] suggested a dynamic rather than fixed location, with two preferred orientations (Fig. 2b, c; Supplementary Fig. 3f). In one orientation, Tyr35[ISCU2] stacks against Trp97[NFS1'] (indicated as 1* in Fig. 2b, c), while in the other (2*) Tyr35[ISCU2] folds back onto ISCU2 and points toward Ser97[ISCU2] and Lys135[ISCU2] with contacts to Glu364[NFS1]. Importantly, the mobility of Tyr35[ISCU2] was observed in multiple crystals of $(NIAU2)_2$ all diffracting to 1.6–1.8 Å resolution. The observed dynamic nature of Tyr35[ISCU2] is at certain variance with the model derived from the cryo-EM map of the $(NIAU2F)_2$ complex, where Tyr35[ISCU2] was modeled only in the pi stacking conformation with Trp97[NFS1'][30]. Re-examination of the ED map (EMD-0560) clearly showed a second orientation of Tyr35[ISCU2] (Supplementary Fig. 4) corroborating our observations with the high resolution X-ray structures.

To explore the contribution of N-terminal residues to the mobility of the mature ISCU2 N terminus, we created a mutant protein in which Y35HKK (representing wild-type ISCU2) was replaced by the LSTQ sequence of ISCU1 (termed ISCU2-L35; Supplementary Fig. 1c), and we determined the structure of $(NIAU2-L35)_2$ (Supplementary Fig. 5a; Supplementary Table 2). Here, only the sidechain of Leu35 was poorly defined in the electron density while Ser36 and the following residues showed an overall well-defined electron density (Fig. 2c, left and middle, Supplementary Fig. 5c), indicating less overall mobility of the N-terminus. Replacing Ser36 in ISCU2-L35 to His present in ISCU2 (creating ISCU2-L35H36; Supplementary Fig. 1c) had a destabilizing effect on the N terminus in the solved $(NIAU2-L35H36)_2$ crystal structure (Fig. 2c, right; Supplementary Fig. 5b, c; Supplementary Table 2), suggesting that the His36[ISCU2] sidechain contributes to the mobility of the ISCU2 N terminus. In conclusion, the high-resolution 3D architecture of $(NIAU2)_2$ complexes shows a flexible N-terminal region, yet otherwise is highly similar to the $(NIAU1)_2$ and $(NIAU2F)_2$ structures.

**Testing the role of pi stacking between Tyr35[ISCU2] and Trp97[NFS1].** From the structural information, we reasoned that pi stacking between Tyr35[ISCU2] and Trp97[NFS1'] may mechanistically explain the importance of Tyr35[ISCU2] (cf. Fig. 2b). Residue Trp97[NFS1] is well conserved in eukaryotic NFS1, but not in prokaryotic IscS. In keeping, in the bacterial IscS-IscU structure the N-terminus of IscU is not in the proximity of the Trp97 equivalent, but loops back to a Met within IscU[46]. We analyzed the impact of exchanging Trp97[NFS1] to Lys, Asp or Ala (NFS1-W97K, -W97D and -W97A mutant proteins) on several biochemical functions of the core ISC complex. Purification of the respective mutant $(NIA)_2$ complexes from $E.~coli$ yielded amounts comparable to wild-type $(NIA)_2$ with similar PLP contents (Supplementary Fig. 6a, b). The affinity of the positively charged NFS1-W97K (as part of $(NIA)_2$) for ISCU2 (determined by microscale thermophoresis) was comparable to wild-type $(NIA)_2$, and that of the negatively charged NFS1-W97D was only twofold lower, i.e., biologically not significant (Fig. 3a; Supplementary Fig. 6c). We then analyzed the NFS1-W97 mutant cysteine desulfurase activities in the presence or absence of FXN, by following the (non-physiological) production of free sulfide in the presence of DTT. As reported earlier[33,35], the simultaneous presence of ISCU2 and FXN increased the NFS1-dependent sulfide release seven-fold (Fig. 3b). The corresponding desulfurase activities of the various W97 mutant $(NIA)_2$ complexes were hardly changed compared to wild-type $(NIA)_2$. Together, these results did not indicate a contribution of Tyr35[ISCU2]-Trp97[NFS1'] pi stacking for $(NIA)_2$ function and ISCU2 interaction.

Further, we investigated the activity of the NFS1-W97 mutant proteins in transferring the persulfide moiety to ISCU2. A thiol alkylation-based assay was used which determines the ISCU2 persulfide modification by a gel shift assay (see cartoon in Supplementary Fig. 7a)[35]. The band-shift of fully maleimide-polyethylene-glycol_{11}-biotin (MPB) modified ISCU2 (4-MPB; reaction in the absence of $(NIA)_2$ or Cys) to the lower molecular

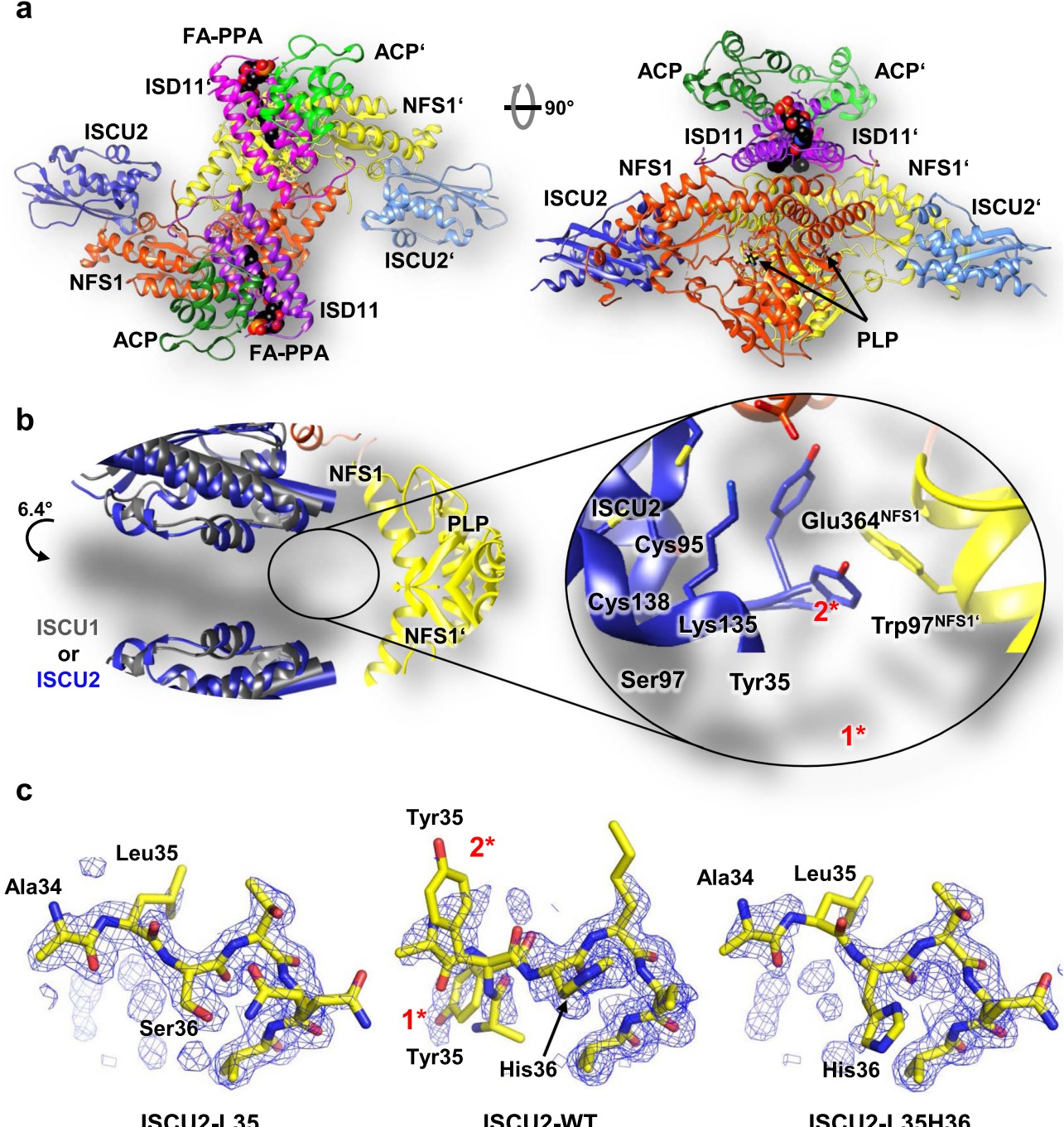

**Fig. 2 High resolution crystal structures of (NIAU2)$_2$ complexes reveal flexibility of the ISCU2 N terminus. a** Overall architecture of the (NIAU2)$_2$ complex obtained by X-ray crystallography. NFS1 subunits are depicted in orange and yellow, ISD11 in purple and magenta, ACP in dark and light green, and ISCU2 in dark and light blue. Each NFS1 subunit covalently binds a pyridoxal phosphate (PLP), and ACP carries a fatty acyl phosphopantetheinyl (FA-PPA) moiety. **b** Detailed view of the 6° tilting of ISCU2 in (NIAU2)$_2$ relative to ISCU1 in (NIAU1)$_2$[28]. The conserved N-terminal Tyr35 of ISCU2 shows a dual localization within the crystal (inset). In one conformation (1*) Tyr35 stacks to Trp97 of NFS1′, and in the other (2*) it contacts Glu364 of NFS1. **c** Effect of exchanges of Tyr35 and His36 of ISCU2 on the mobility of the N terminus (for detailed crystallographic statistics refer to Supplementary Table 2). Residues 34–39 of ISCU2 (mutants) are shown by sticks, and water molecules were omitted. The electron density 'omit' map 2Fo-DFc is displayed at 0.8 sigma level near the N terminus of ISCU2. The map was calculated after removing residues 34–38 from the model and refining the model for one cycle with Phenix software. For stereo views refer to Supplementary Fig. 5c. Left: The mutant protein ISCU2-L35 shows a well-ordered N terminus with well-defined electron density for all N-terminal residues (the density for the Leu sidechain is somewhat lower, suggesting increased mobility relative to the backbone); middle: wild-type ISCU2 shows poor electron density at the N terminus indicating high mobility, and was modeled with two possible conformations for Tyr35; right: The mutant protein ISCU2-L35H36 shows poorer electron density for the N terminus indicating that the His36 sidechain increases mobility.

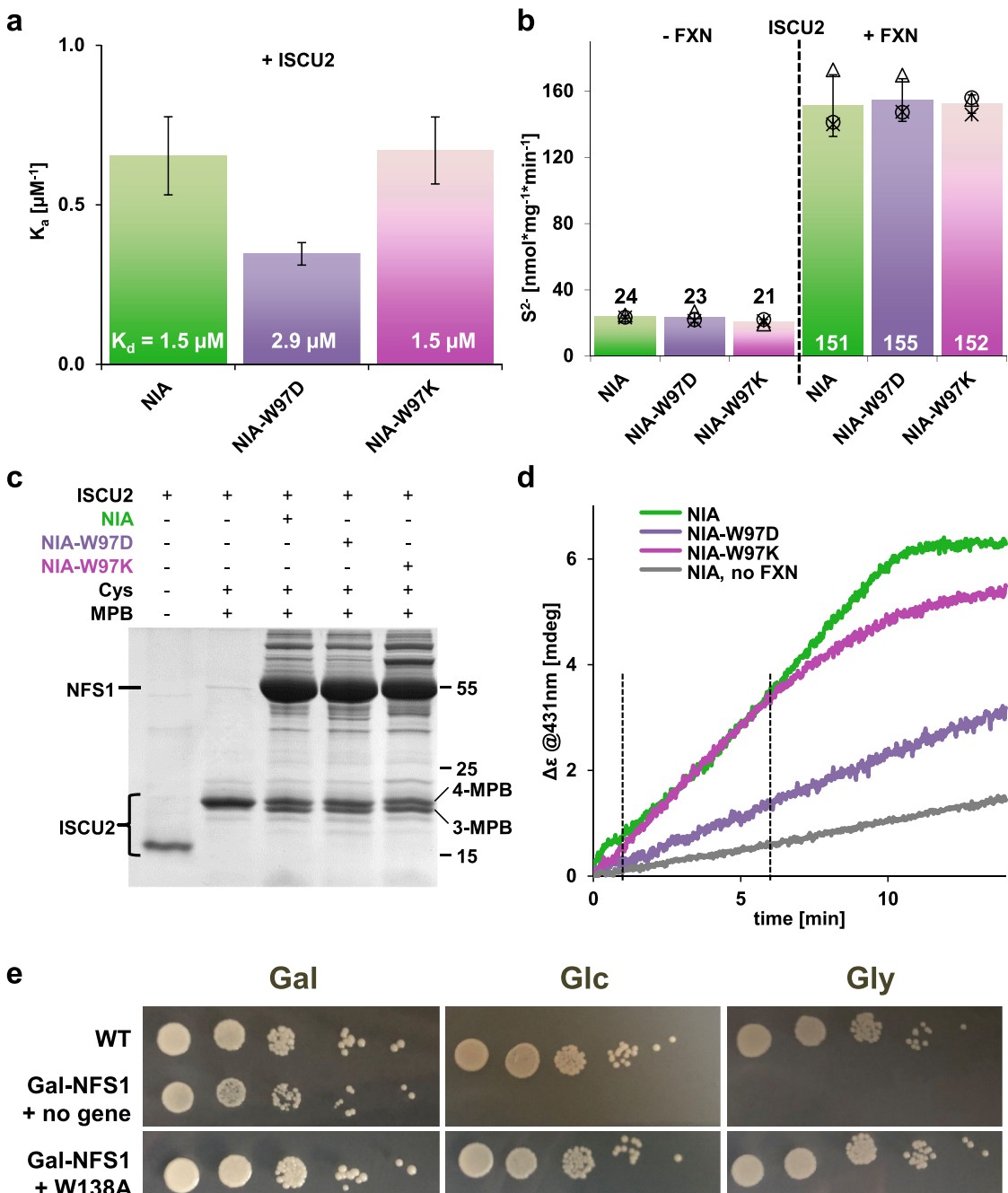

**Fig. 3 No crucial role of Trp97 of NFS1 in early steps of core ISC complex function. a** Affinities of NFS1 or indicated NFS1-W97 mutant proteins (as part of purified (NIA)$_2$) for ISCU2 were estimated by microscale thermophoresis. The numbers within the bars represent $K_d$ values (error bars represent SD of the fit using three independent biological replicates, Supplementary Fig. 6c). **b** Determination of DTT-mediated sulfide (S$^{2-}$) production by the cysteine desulfurase NFS1 or indicated mutant proteins in the presence of ISCU2 without and with FXN ($n = 3$ biological replicates, individual data points are shown, error bars show SD). **c** Persulfide transfer from NFS1 or the indicated NFS1-W97 mutant proteins (with added FXN) onto ISCU2 was assayed by Cys alkylation with maleimide-polyethylene-glycol$_{11}$-biotin (MPB) as described in Supplementary Fig. 7a. Samples were analyzed by SDS-PAGE. As controls, the running behavior of ISCU2 minus and plus MPB alkylation and Cys is shown in the first two lanes. **d** Representative data for enzymatic Fe/S cluster reconstitution on ISCU2 with NFS1 or indicated NFS1-W97 mutant proteins. As a control, FXN was omitted for the wild-type reaction (no FXN). Dashed lines indicate the period used for initial rate determination (Supplementary Fig. 7b). ($n = 3$ biological replicates) **e** Gal-NFS1 yeast cells were transformed with vectors (low copy, native promoter) encoding either wild-type Nfs1 (WT), no protein, or Nfs1-W138A mutant protein (yeast Nfs1-W138 corresponds to human NFS1-W97). Cells were grown on glucose for 24 h to deplete endogenous Nfs1, and were subsequently spotted onto SC agarose plates containing galactose (Gal) to promote expression of chromosomal *NFS1* or glucose (Glc) or glycerol (Gly) to repress *NFS1* expression. A fivefold serial dilution of cells was grown for 3 days at 30 °C. Source Data are provided as a Source Data file.

mass of (NIA)$_2$- and Cys-treated ISCU2 (3-MPB) indicated the persulfide transfer, which occurred at similar efficiencies for wild-type NFS1 and both NFS1-W97 mutant proteins showing no influence of this residue for this reaction (Fig. 3c). Finally, we tested the ability of the NFS1-W97 mutant proteins to de novo assemble a [2Fe-2S] cluster on ISCU2 by enzymatic reconstitution[25,28,47]. In this CD spectroscopy-based assay, physiological [2Fe-2S] cluster formation on ISCU2 dimers is followed under anaerobic conditions, and depends on the (NIA)$_2$ complex, FXN, and the electron transfer chain NADPH-FDXR-FDX2, plus Fe$^{2+}$ and Cys. The rates of [2Fe-2S] formation by the NFS1-W97 mutant proteins were either not perturbed (W97K) or decreased 2.3-fold (W97D) relative to wild-type NFS1 (Fig. 3d; Supplementary Fig. 7b). The possibility that the lysine in NFS1-W97K may form a cation-pi interaction with Tyr35$^{ISCU2}$ was ruled out by examining the NFS1-W97A variant showing almost wild-type activity (Supplementary Fig. 7c). The unchanged activities of NFS1-W97K and NFS1-W97A in [2Fe-2S] cluster synthesis on ISCU2 excluded a crucial contribution of Trp97$^{NFS1}$ to core ISC complex function. This conclusion was further supported by an in vivo growth test using the galactose-inducible yeast strain Gal-NFS1 in which Nfs1 can be depleted by growth on glucose or glycerol[48]. The resulting growth defect could be fully restored by expressing the yeast Nfs1-W138A mutant protein carrying a site-specific exchange of Trp138 (corresponding to Trp97$^{NFS1}$) to Ala suggesting that this mutant protein was functional (Fig. 3e). Collectively, these results did not indicate any critical role of pi stacking between Tyr35$^{ISCU2}$ and Trp97$^{NFS1'}$ despite their conservation.

**Tyr35$^{ISCU2}$ is dispensable in early steps of de novo [2Fe-2S] cluster synthesis.** The insignificant contribution of pi stacking between Tyr35$^{ISCU2}$ and Trp97$^{NFS1}$ leaves open at which step of [2Fe-2S] cluster synthesis the essential Tyr35$^{ISCU2}$ becomes crucial. We compared the binding affinities of the (NIA)$_2$ complex to ISCU2 with those to ISCU2-Y35A, ISCU2-L35, and ISCU2-L35H36, yet found no significant differences (Fig. 4a; Supplementary Fig. 8a). We then compared ISCU2 with the three variants in their abilities to stimulate the cysteine desulfurase activity plus or minus FXN (cf. Figure 3b). No significant differences were observed in the amounts of DTT-dependent sulfide generation (Fig. 4b). Next, we measured the ability of ISCU2 or the respective variants to receive the persulfide from NFS1 using the thiol alkylation-based assay (cf. Supplementary Fig. 7a). Similar efficiencies of persulfide transfer from NFS1 to all tested ISCU2 proteins were obtained (Fig. 4c). Together, these results exclude a crucial function of Tyr35$^{ISCU2}$ in the initial steps of core ISC complex function. In contrast, no [2Fe-2S] cluster could be enzymatically reconstituted on ISCU2-Y35A, ISCU2-L35 or ISCU2-L35H36; Fig. 4d). However, these proteins could still assemble wild-type amounts of [2Fe-2S] clusters by chemical reconstitution demonstrating their structural integrity (Supplementary Fig. 8b). This striking difference between enzymatic and chemical [2Fe-2S] cluster reconstitution indicates a crucial role of the universally conserved Tyr35$^{ISCU2}$ during physiological [2Fe-2S] cluster synthesis (cf. Fig. 1). Our data suggest that Tyr35$^{ISCU2}$ is not required during early steps of core ISC complex function, yet rather performs its essential physiological function after persulfide transfer to ISCU2.

**Tyr35$^{ISCU2}$ dimerization induces [2Fe-2S] cluster synthesis.** The reactions supported by the core ISC complex after persulfide transfer to ISCU2 are mechanistically poorly understood. They include the FDX2-mediated reduction of the persulfide sulfur (S$^0$) to sulfide (S$^{2-}$) present in Fe/S clusters, the synthesis of the [2Fe-

2S] cluster, the dimerization of ISCU2 to coordinate the bridging [2Fe-2S] cluster, and the ISCU2 dissociation from the core ISC complex. An effect of Tyr35$^{ISCU2}$ on FDX2-mediated persulfide reduction was ruled out by adding the electron transfer chain (FDX2, FDXR, NADPH) to the thiol alkylation-based assay after persulfidation of ISCU2[36]. Reductive cleavage of the ISCU2 persulfide by FDX2 was highly efficient for both ISCU2 and ISCU2-Y35A (Fig. 5a). Additionally, from its location distant to the active center of ISCU2, Tyr35 is unlikely to perform a direct role in Fe/S cluster synthesis. Based on the flexibility of the ISCU2 N-terminus (see above), we hypothesized that two Tyr35$^{ISCU2}$ bound to different ISC core complexes could interact, thereby inducing ISCU2 dimerization, concomitant [2Fe-2S] cluster synthesis from ISCU2-bound iron and sulfide, and dissociation from (NIA)$_2$. To test this idea, we asked whether the putative Tyr35-Tyr35 interaction could be functionally replaced by an ion pair interaction. We produced recombinant mature ISCU2-Y35K and ISCU2-Y35D mutant proteins, and investigated their activities. When tested individually, both mutant proteins were fully active in the initial steps of core ISC function as evident from wild-type efficiencies in receiving the persulfide modification (Fig. 5b; cf. Fig. 3c). Conversely, they did not support any [2Fe-2S] cluster assembly by enzymatic reconstitution, despite the presence of Fe and cysteine (Fig. 5c). The findings are consistent with the important late function of Tyr35$^{ISCU2}$. Strikingly, when the two mutant proteins were added simultaneously during enzymatic reconstitution, [2Fe-2S] cluster formation was observed at a slightly decreased rate and 50% efficiency compared to wild-type ISCU2. Doubling the amount of the ISCU2 mutant proteins resulted in wild-type yields of holo-formation. The resulting CD spectrum was virtually identical to that of wild-type ISCU2 indicating the generation of a canonical [2Fe-2S] cluster (Supplementary Fig. 9a). In conclusion, the essential N-terminal Tyr35$^{ISCU2}$ can be replaced by oppositely charged residues in two different ISCU2 proteins, supporting the idea of a decisive transient interaction of two Tyr35 during physiological [2Fe-2S] cluster synthesis.

We next tested the requirement of Tyr35 for ISCU2 dimerization. The various enzymatic reconstitution mixtures from above were subjected to anaerobic analytical gel filtration, and the oligomeric state of the ISCU2 proteins (20-fold excess over the other core ISC proteins) was recorded by multi-wavelength UV/Vis spectroscopy to determine ISCU2 protein (280 nm) or bound Fe/S cluster (320 and 420 nm). Without enzymatic reconstitution (no Fe/no NADPH), both wild-type ISCU2 and the ISCU2-Y35K + ISCU2-Y35D mixture eluted as monomers with no detectable Fe/S cluster (Fig. 5d, Supplementary Fig. 9b). Likewise, individually reconstituted ISCU2-Y35K or ISCU2-Y35D remained monomeric with no indication of Fe/S cluster formation. In contrast, virtually all of enzymatically reconstituted wild-type ISCU2 and ca. 50% of the ISCU2-Y35K + ISCU2-Y35D mixture eluted as dimers with bound Fe/S cluster. In summary, these results clearly indicated that both [2Fe-2S] cluster formation and ISCU2 dimerization crucially depends on Tyr35$^{ISCU2}$, yet this hydrophobic residue can be replaced efficiently by ISCU2 with two oppositely charged residues, even though these mutant proteins per se are inactive. The findings provide a satisfactory mechanistic explanation for the critical role of N-terminal Tyr35$^{ISCU2}$ in Fe/S protein biogenesis and its universal conservation in ISCU2-like proteins.

Apparently, during the final steps of core ISC complex function two Tyr35 of different ISCU2 molecules directly interact to generate the ISCU2 dimer and to form the [2Fe-2S] cluster. The Tyr-Tyr interaction could be directly visualized by a UV-crosslinking approach in which covalent di-tyrosine adducts are generated[49]. UV irradiation of ISCU2, but not of ISCU2-Y35D

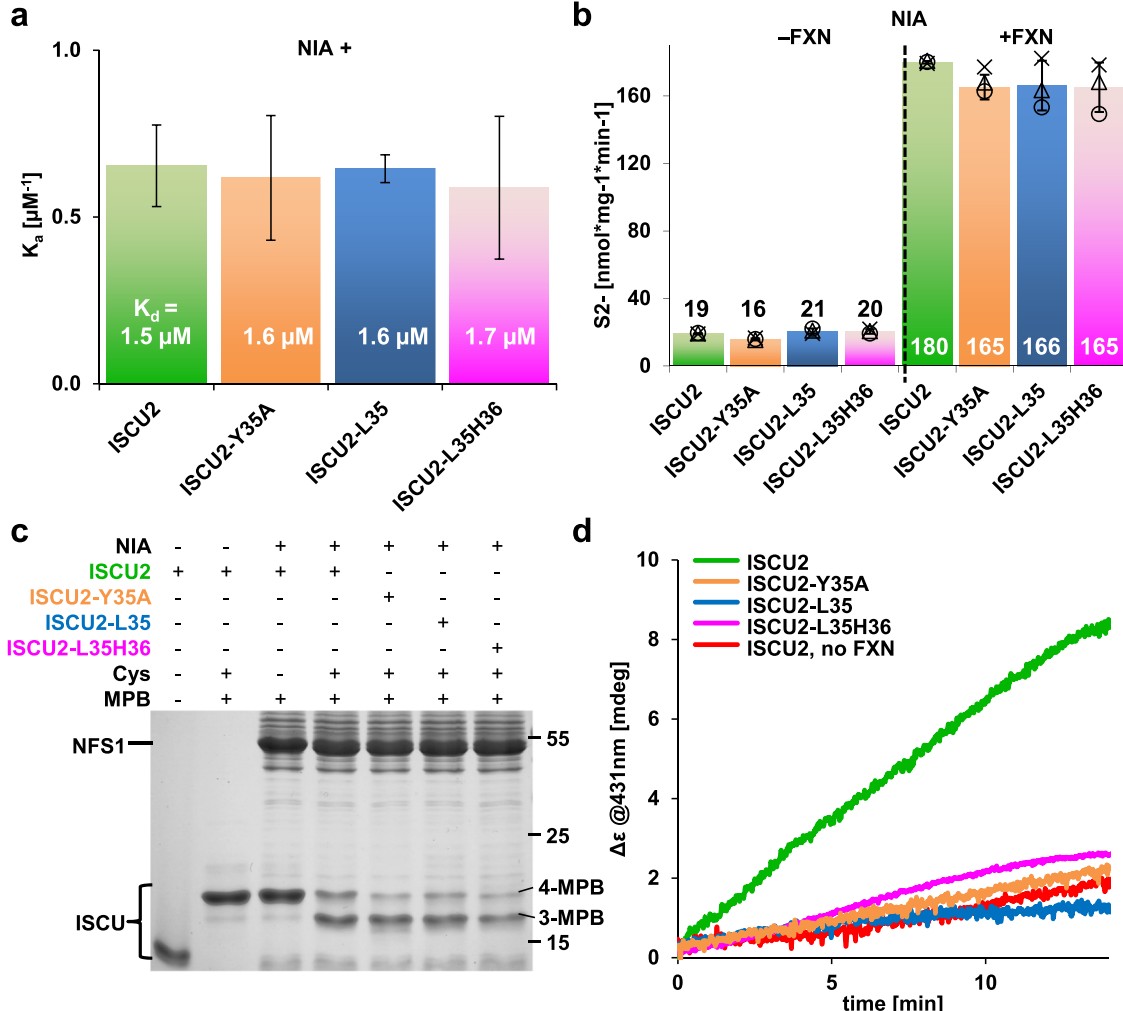

**Fig. 4 Crucial role of Tyr35 of ISCU2 in late but not early steps of core ISC complex function. a** Affinities of the indicated ISCU2 proteins for wild-type $(NIA)_2$ were measured by microscale thermophoresis (the data for wild-type ISCU2 was taken from Fig. 3a). The numbers within the bars represent $K_d$ values (error bars represent SD of the fit using three independent experiments, Supplementary Fig. 8a). **b** Determination of DTT-mediated sulfide ($S^{2-}$) production by the $(NIA)_2$ complex in the presence of the indicated ISCU2 proteins without and with FXN after 20 min of incubation ($n = 3$ biological replicates, individual data points are shown, error bars show SD). **c** Persulfide transfer from NFS1 onto the indicated ISCU2 proteins was assayed as in Fig. 3c. **d** Enzymatic [2Fe-2S] cluster reconstitution (cf. Fig. 3d) of the indicated ISCU2 proteins with wild-type $(NIA)_2$. Source Data are provided as a Source Data file.

resulted in characteristic fluorescence changes indicative of the formation of di-tyrosine (maximal fluorescence emission between 380 and 460 nm[49]) (Fig. 6a). The UV-induced di-tyrosine crosslink was also evident from SDS-PAGE showing efficient dimer formation for wild-type ISCU2, but not for ISCU2-Y35D excluding that other Tyr residues gave rise to crosslinking (Fig. 6a, inset). Together, our findings demonstrate that the interaction of two Tyr35[ISCU2] residues in two ISCU2 proteins is essential for both dimerization and [2Fe-2S] cluster formation.

To finally assess the question whether two ISCU2 bound to different $(NIA)_2$ complexes or one $(NIA)_2$-bound ISCU2 plus one free ISCU2 give rise to the final dimeric product, we first bound equimolar HIS-tagged ISCU2 to $(NIA)_2$ in the presence of FXN and FDX2 (for experimental setup see Supplementary Fig. 10a). We then added free cysteine (to start the reaction) and a twofold excess of non-tagged ISCU2 (as a source for free ISCU2 in solution), and performed the enzymatic [2Fe-2S] cluster reconstitution followed by aSEC. The resulting dimeric holo-ISCU2 product exclusively contained HIS-tagged ISCU2 as shown by

SDS-PAGE and Western blot of the gel filtration samples containing the $(NIA)_2$ complex, ISCU2 dimer and monomer (Fig. 6b, c, Supplementary Fig. 10b). Interestingly, after the reaction about half of the untagged ISCU2 was isolated either with $(NIA)_2$ or as a monomer, indicating that free ISCU2 replaced $(NIA)_2$-bound ISCU2-HIS after dimer formation. This result clearly shows that two ISCU2-HIS bound to different $(NIA)_2$ complexes exclusively formed the holo-ISCU2-HIS dimer, while free apo-ISCU2 was inactive, yet could enter the $(NIA)_2$ complex after dissociation of ISCU2-HIS. Since both Fe and S are present in $(NIA)_2$-bound ISCU2-HIS in our experimental setup[25,36], our findings strongly suggest that the [2Fe-2S] cluster is formed by combining 1Fe-1S on different $(NIA)_2$-bound ISCU2 monomers during Tyr35-mediated dimerization (Fig. 7). Collectively, enzymatic reconstitution, gel filtration and cross-linking approaches employing wild-type and different mutant ISCU2 proteins revealed the essential mechanistic role of Tyr35 in triggering ISCU2 dimerization and concomitant [2Fe-2S] cluster synthesis.

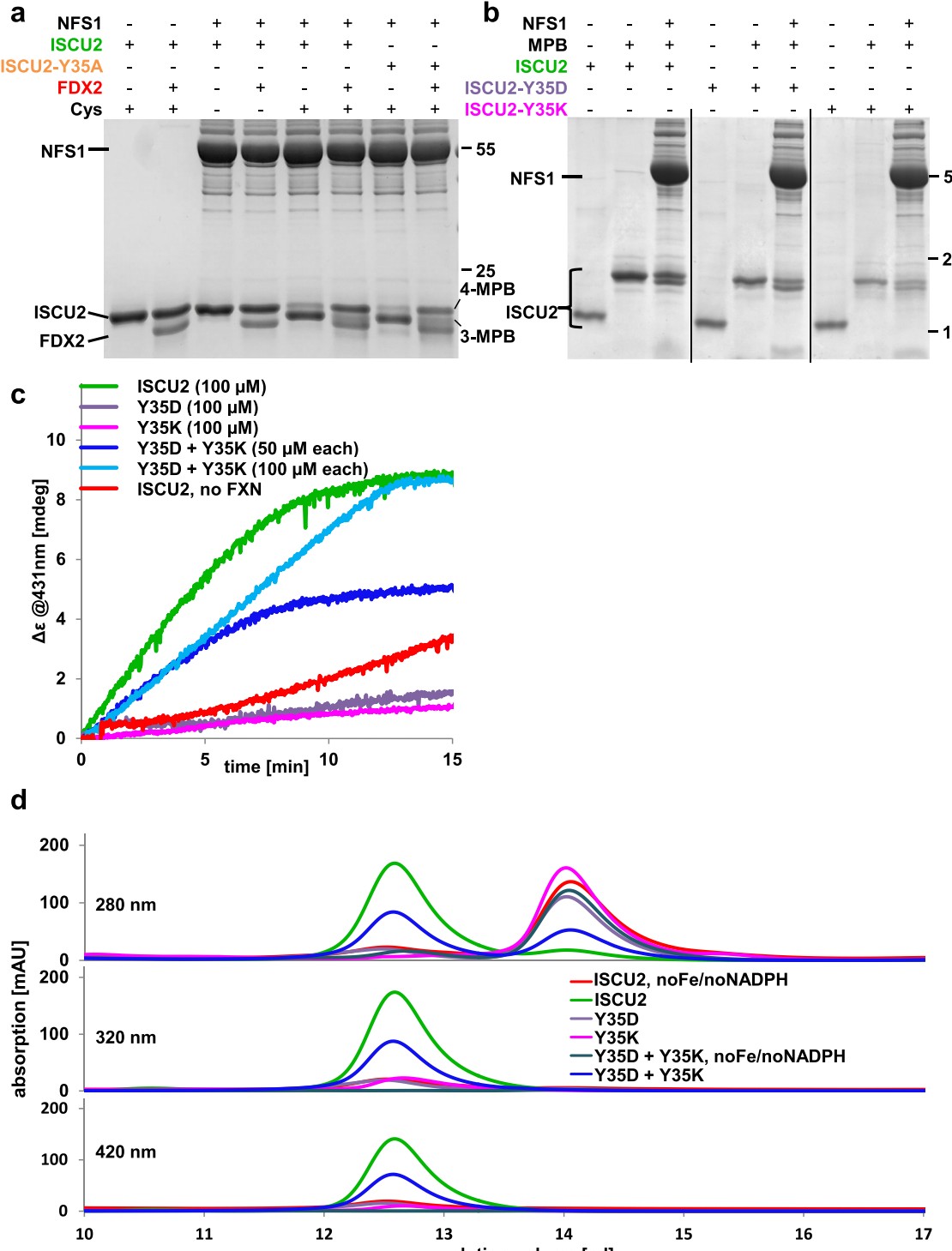

**Fig. 5 The N-terminal Tyr35 of ISCU2 is not required for persulfide reduction, yet essential for [2Fe-2S] cluster synthesis and ISCU2 dimerization.**
**a** Reduction of ISCU2-Y35A-bound persulfide by FDX2 assayed as described in Fig. 3c (see also Supplementary Fig. 7a). Reactions without (NIA)$_2$ or cysteine served as controls (lanes 1 + 2 and 3 + 4, respectively). The re-appearance of fourfold MPB-labeled ISCU2 in lanes 6 and 8 indicates efficient persulfide reduction by FDX2. The experiment was repeated twice with similar outcome. **b** Persulfide transfer from NFS1 onto wild-type ISCU2 as well as ISCU2-Y35D or ISCU2-Y35K mutant proteins. Reactions without (NIA)$_2$ plus and minus MPB served as controls. The experiment was repeated at least three times with similar outcome. **c** Enzymatic reconstitution (cf. Fig. 3d) of wild-type ISCU2 or the ISCU2 mutant proteins Y35D, Y35K, and the mixture of Y35D and Y35K as indicated. **d** Anaerobic analytical size-exclusion chromatography (aSEC) of enzymatically reconstituted ISCU2 and variants (color coding as in **b**). The majority of ISCU2 and ca. 50% of the Y35D + Y35K mixture form dimers that carry a [2Fe-2S] cluster as indicated by absorption at 320 nm and 420 nm and CD spectrum (Supplementary Fig. 9a). In contrast, ISCU2-Y35D and ISCU2-Y35K eluted as monomers like apo-ISCU2, and showed no absorbance at 320 nm or 420 nm indicating that no Fe/S cluster was formed. Source Data are provided as a Source Data file.

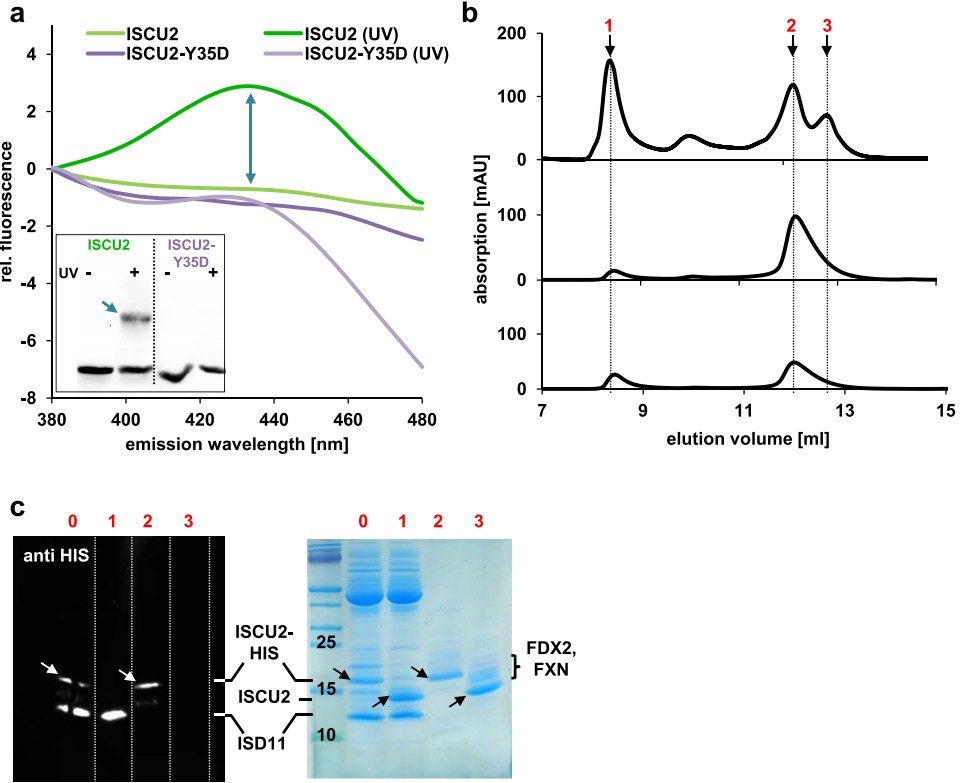

**Fig. 6 Tyr35-Tyr35 interactions between two (NIA)₂-bound ISCU2 are crucial for triggering [2Fe-2S] cluster formation by ISCU2 dimerization.**
**a** Di-tyrosine formation by UV-crosslinking. Chemically reconstituted ISCU2 but not ISCU2-Y35D (lacking N-terminal Tyr35, but having two other Tyr) show a fluorescence increase (arrow) at $\lambda_{Ex}$ 320 nm and $\lambda_{Em}$ 380–460 nm upon UV irradiation characteristic for di-tyrosine formation. The spectra recorded before and after crosslinking are set to the values at $\lambda_{Em}$ 380 nm. The inset shows an immunoblot of a reducing SDS-PAGE from the UV-treated samples stained for the HIS-tag of the ISCU2 proteins. The band marked by the arrow represents the di-tyrosine-linked ISCU2 dimer. The experiment was repeated twice with similar outcome. **b** Only (NIA)₂-bound ISCU2 and not free ISCU2 gives rise to holo-ISCU2 dimer formation. For experimental setup see Supplementary Fig. 10a. Representative aSEC profiles are shown for absorption at 280 nm (protein), 320 nm (Fe/S cluster) and 420 nm (Fe/S cluster and PLP). **c** Fractions from peaks 1 to 3 of part b containing (NIA)₂, dimeric ISCU2-HIS and monomeric ISCU2, respectively, were analyzed by SDS-PAGE and subsequent anti-HIS tag immunostaining or Coomassie staining. The reaction mixture prior to Cys addition (sample 0) served as a control (cf. Supplementary Fig. 10a). The experiment was repeated twice with similar outcome. Source Data are provided as a Source Data file.

## Discussion

Mitochondrial de novo [2Fe-2S] cluster synthesis is the initial step of cellular Fe/S protein biogenesis, and, not surprisingly, this key biochemical reaction is essential for life. Previous in vivo studies and the in vitro reconstitution of this reaction have revealed the cooperative requirement and function of seven ISC proteins[21,25,28,36]. The cysteine desulfurase (NIA)₂ complex (NFS1-ISD11-ACP) releases sulfur from free cysteine to form a protein-bound persulfide intermediate, which is then transferred in a FXN-dependent fashion to the iron-binding scaffold protein ISCU2[32–35]. The redox chain of NADPH-FDXR-FDX2 reduces the persulfide to sulfide, eventually leading to a bridging [2Fe-2S] cluster bound to dimeric ISCU2. Here, we have concentrated on the mechanistic elucidation of the least understood, final part of this multi-step reaction, namely the actual [2Fe-2S] cluster formation. So far, it was unknown how the ISCU2-bound iron and persulfide are converted into the [2Fe-2S] cluster, and several hypothetical models have been suggested[50]. We now provide biochemical, spectroscopic, and structural evidence for the interaction of two N-terminal Tyr35$^{ISCU2}$ residues with mechanistic importance for both ISCU2 dimerization and concomitant [2Fe-2S] cluster formation. Tyr35$^{ISCU2}$ is strictly conserved in evolution, and is essential for viability of yeast, E. coli, and man[41,42] (this study). Here, we found that this residue can be replaced functionally by two oppositely charged residues present in two different (NIA)₂ complex-bound ISCU2 mutant proteins.

Individually, these ISCU2 variants do not support any [2Fe-2S] cluster formation but in combination show almost wild-type activity. The findings demonstrate that hydrophobic Tyr35-Tyr35 interactions, directly evidenced by UV-crosslinking, can efficiently be replaced by ionic forces to support both ISCU2 dimerization and [2Fe-2S] cluster formation. We also showed that only (NIA)₂-bound ISCU2, and not free ISCU2, can give rise to the holo-ISCU2 dimer. We therefore propose that [2Fe-2S] cluster synthesis is the result of Tyr35-Tyr35 interaction-triggered dimerization of two 1Fe-1S-binding ISCU2 attached to different (NIA)₂ complexes (Fig. 7).

Our mechanistic studies of [2Fe-2S] cluster synthesis on ISCU2 and the identification of the mechanistic role of Tyr35$^{ISCU2}$ were guided by new high resolution X-ray structures (best resolution of 1.57 Å) of various (NIAU2)₂ complexes with clearly resolved sidechain conformations. These structures show a substantially better resolution compared to our X-ray structures of (NIA)₂-containing cytosolic ISCU1 or to the cryo-EM structure of (NIAU2F)₂ (both around 3.2 Å), even though the overall structures of all complexes and their individual components are similar[28,30]. The high resolution (NIAU2)₂ structures revealed a conspicuous flexibility of the N-terminal region of ISCU2 with Tyr35$^{ISCU2}$ being present in two main conformations. In one of the modeled conformations, Tyr35$^{ISCU2}$ undergoes hydrogen bonding with Glu364 of NFS1, while in the other it makes a hydrophobic contact with Trp97 of NFS1'. Similar flexibility is

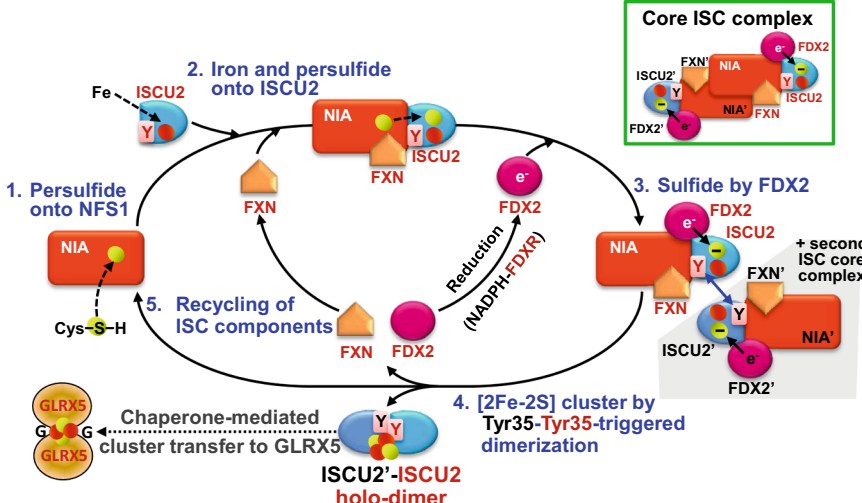

**Fig. 7 Mechanistic working model for the reaction cycle of [2Fe-2S] cluster formation on an ISCU2 dimer by the mitochondrial core ISC complex.** For simplicity, only one half of the symmetrical core ISC complex (green box, upper right) is shown in the reaction cycle. The steps are: 1. A persulfide is generated in a PLP-dependent fashion on a flexible Cys-containing loop of NFS1 (as part of the (NIA)₂ complex). 2. Upon binding of Fe-loaded ISCU2 and FXN to NFS1, the persulfide is rapidly transferred from NFS1 to an active-site Cys of ISCU2. 3. Binding of reduced FDX2 assists the reduction of persulfide sulfur to sulfide (-). 4. Tyr35-mediated dimerization of two (Fe and S ion-binding) ISCU2 proteins on different core ISC complexes yields a bridging [2Fe-2S] cluster on an ISCU2 dimer. 5. Finally, the [2Fe-2S] cluster is further transferred to glutaredoxin 5 (GLRX5) in chaperone-dependent fashion, and (NIA)₂, FXN, oxidized FDX2, and apo-ISCU2 are recycled for the next synthesis round. Red and yellow circles, iron and sulfur; Y, Tyr35; e⁻, reducing electron on FDX2.

also evident in the cryo-EM map of the (NIAU2F)₂ complex. Since it is less clear than in the X-ray structure, it was not modeled[30]. To structurally examine the reason for the N-terminal flexibility further, two additional (NIAU2)₂ complexes containing the ISCU2-L35 and ISCU2-L35H36 N-terminal mutant proteins were crystallized (Supplementary Fig. 1c). Comparison of these structures showed a remarkable influence of both Tyr35ISCU2 and His36ISCU2 on the flexibility of the N terminus of ISCU2. While ISCU2-L35 showed well-defined electron density for the N-terminal residues (with B-factors ~40–50), re-introduction of His36ISCU2 in ISCU2-L35H36 destabilized its N terminus (weaker electron density, B-factors 65–80). In comparison, the N-terminal region of wild-type ISCU2 within (NIAU2)₂ showed an even higher conformational flexibility. In all these structures, residue 37 (Lys or Thr) and the subsequent ones were well-defined (B-factors of ~30–40). The structural flexibility of the ISCU2 N terminus is likely crucial for the dynamic Tyr-Tyr interactions during [2Fe-2S] cluster formation and dimerization.

The vicinity of the flexible Tyr35ISCU2 and Trp97NFS1 in the (NIAU2)₂ structures does not appear to be of functional importance. Various mutant NFS1-W97 proteins showed no drastic changes in, e.g., the affinities of binding to ISCU2 or in sulfide production by the addition of ISCU2 and FXN[33,35]. Further, they could efficiently transfer persulfide to ISCU2 and were not impaired in enzymatic Fe/S cluster reconstitution indicating no critical role of Trp97NFS1 in these steps of Fe/S cluster formation. Vice versa, mutation of Tyr35ISCU2 to Ala did not impair ISCU2 interaction with (NIA)₂ and persulfide transfer. Together, these biochemical results render a critical contribution of Trp97NFS1-Tyr35ISCU2 contacts to NFS1-ISCU2 interaction and function unlikely. This view was fully supported in vivo showing normal growth of the corresponding yeast mutant Nfs1-W138A.

In contrast to Trp97NFS1, the N-terminal Tyr35ISCU2 was found to be crucial for cellular Fe/S protein biogenesis in human cells. Biochemical dissection of the individual steps of mitochondrial [2Fe-2S] cluster synthesis showed that Tyr35ISCU2 is dispensable for the initial steps, up to and including persulfide

transfer from NFS1 to ISCU2 and the subsequent FDX2-mediated persulfide reduction to sulfide. In contrast, exchanges of Tyr35 in ISCU2-Y35A, ISCU2-L35 and ISCU2-L35H36 did not support any [2Fe-2S] cluster assembly by enzymatic reconstitution. Surprisingly, these proteins were still able of assembling a [2Fe-2S] cluster by chemical reconstitution showing that in principle they are capable of binding a Fe/S cluster. Nevertheless, these mutant proteins were crippled in assembling the [2Fe-2S] cluster via the physiological pathway demonstrating that Tyr35ISCU2 is required only in the later phase of [2Fe-2S] cluster synthesis, i.e., after persulfide transfer to iron-loaded ISCU2 and its reduction to sulfide. These reactions include the de novo synthesis of the [2Fe-2S] cluster, dimerization of ISCU2 to coordinate the bridging [2Fe-2S] cluster, and ISCU2 dissociation from the core ISC complex[25,36].

Since all N-terminal ISCU2-Y35 mutant proteins were unable to assemble a [2Fe-2S] cluster, we reasoned that the physiological role of Tyr35ISCU2 may be in facilitating the dimerization of ISCU2 proteins each holding one iron and one sulfur ion. A confirmation of this idea came from the use of two ISCU2 proteins in which Tyr35ISCU2 was exchanged to oppositely charged residues to replace the suspected hydrophobic Tyr35-Tyr35 contact by an ion pair interaction. ISCU2-Y35D and ISCU2-Y35K were able to receive a persulfide from NFS1 as efficiently as wild-type ISCU2, yet individually did not support enzymatic [2Fe-2S] cluster synthesis and dimerization, as observed by CD spectroscopy and analytical gel-filtration with multi-wavelength detection. In striking contrast, co-incubation of the two variants restored [2Fe-2S] cluster formation on an ISCU2 dimer at wild-type efficiency, even though at a slightly lower rate. The close vicinity of the N-terminal Tyr35 within a holo-ISCU2 dimer was directly shown by UV-induced crosslinking and detection of the di-Tyr product by fluorescence as well as of the crosslinked ISCU2 dimer by gel electrophoresis. As a control, the ISCU2-Y35D mutant protein did not undergo any crosslink, despite the presence of two other, partially surface-exposed Tyr residues in ISCU2-Y35D. Interestingly, during revision of this work an X-ray

structure of holo-IscU from the Euryarchaeon *Methanotrix* (also encoding SufSBC proteins) was published[51]. In this structure, a [2Fe-2S] cluster is associated with an IscU monomer, and none of the different IscU dimers within the structure carry their N-terminal Tyr residues in close vicinity, indicating clear differences of this archaeal IscU structure to our physiological findings for human ISCU2. Notably, the positions of the N-terminal Tyr residues in the IscU structure do not readily explain their essential function. In summary, our studies unequivocally identify a crucial role of the N-terminal Tyr35$^{ISCU2}$ residue in triggering dimerization and [2Fe-2S] cluster formation by direct interaction. Since the N-terminal Tyr of *E. coli* IscU can be replaced by Phe, His and Trp residues, these interactions seem to involve hydrophobic pi stacking[42]. The fact that nature exclusively uses Tyr at this position points to an additional important role of its hydroxyl group.

A mechanistically important problem was the question of whether dimeric holo-ISCU2 is generated from two ISCU2 bound to different (NIA)$_2$ complexes or from one (NIA)$_2$-bound ISCU2 plus one free ISCU2. We have not observed any Fe/S cluster formation on ISCU2, particularly on ISCU2-Y35 mutant proteins, as part of the (NIAU2)$_2$ complex. Therefore, [2Fe-2S] cluster formation on a single core ISC complex-associated ISCU2 seemed unlikely. However, for an archaeal IscS-IscU complex (containing mutated IscU$^{D35A}$) a stably bound subunit-bridging [2Fe-2S] cluster has been documented[39]. To resolve the question for the mitochondrial core ISC system, we first bound equimolar amounts of HIS-tagged ISCU2 to (NIA)$_2$ in the presence of FXN and FDX2, and then started the reaction by adding both free cysteine and excess non-tagged ISCU2. Gel filtration combined with spectroscopic and biochemical analyses clearly indicated that the resulting dimeric holo-ISCU2 product exclusively contained HIS-tagged ISCU2, while untagged ISCU2 was isolated both in complex with (NIA)$_2$ (after ISCU2-HIS dissociation) or as a monomer. Since under the reconstitution conditions (NIA)$_2$-bound ISCU2-HIS contains both Fe and S ions[25,36] (this work), collectively the findings suggest that dimeric holo-ISCU2 is generated by Tyr35-mediated dimerization of two 1Fe-1S ion-containing ISCU2 monomers bound to different (NIA)$_2$ complexes.

Our findings together with numerous published studies (reviewed in[4]) allow us to propose a molecular mechanism for the complete cycle of mitochondrial [2Fe-2S] cluster biogenesis. The reactions start with the PLP-dependent sulfur release from free cysteine by NFS1 to create an enzyme-bound persulfide (step 1 in Fig. 7)[31]. Persulfide transfer from NFS1 to Cys138 of iron-loaded ISCU2 is stimulated by binding of FXN (step 2)[32,36]. The persulfide on ISCU2 is then reduced by electron transfer from ferredoxin FDX2 to create an ISCU2-bound sulfide (step 3)[25,47]. The actual [2Fe-2S] cluster synthesis (step 4) is initiated by Tyr35-Tyr35 interaction that was directly visualized by UV-crosslinking. The 'hydrophobic handshake' of the two flexible N-terminal Tyr35 then triggers ISCU2 dimerization and the generation of a bridging [2Fe-2S] cluster. Importantly, the two 1Fe-1S-containing ISCU2 proteins forming the holo-ISCU2 dimer derive from different (NIAU2)$_2$ complexes, rather than from one (NIA)$_2$-bound ISCU2 and one free ISCU2 (Fig. 7). The final step 5 of the reaction cycle involves the recycling of the individual ISC components including the chaperone-mediated forward transfer of the ISCU2-bound [2Fe-2S] cluster to GLRX5 and late-acting ISC proteins, as well as the reduction of oxidized FDX2 by NADPH-FDXR. Whether the principles of this mitochondrial pathway are also of relevance for bacterial IscU-mediated reactions requires dedicated in vivo and in vitro studies, yet seems attractive to us based on the evolutionary conservation of most core ISC complex components and Tyr35$^{ISCU2}$. While our model describes the full

reaction cycle in reasonable detail, a number of open questions remain to be resolved. For instance, it is unknown whether FXN and FDX2 dissociation from the (NIAU2)$_2$ complex is connected to ISCU2 dimerization and how the resulting [2Fe-2S] cluster is coordinated by dimeric ISCU2. The highly efficient and physiologically relevant enzymatic reconstitution system for de novo [2Fe-2S] cluster synthesis is ideal for addressing these mechanistic problems.

## Methods

**Plasmids.** The RNAi as well as the oligonucleotides used for in vivo studies in human cell culture are listed in Supplementary Table 3. Expression of human ISCU2 proteins in HeLa cells was suppressed by a short hairpin RNA (shRNA) approach[52], based on an efficient siRNA sequence[53] which is directed against both human ISCU2 and ISCU1. A mammalian expression vector containing full-length *ISCU2* cDNA (NM_213595.4) was obtained from Thermo Scientific (United Kingdom), and the ISCU2 ORF was used to substitute the EGFP site within pEGFP-N3 (Clontech, USA). Removal of the nucleotide stretch (base pairs 7–102) encoding the mitochondrial localization sequence (MLS) yielded a plasmid encoding ISCU2ΔMLS (Supplementary Fig. 1c). A site-directed mutagenesis approach[54] was used to introduce a Y35A amino acid substitution into full-length ISCU2. Similarly, silent mutations (sm) corresponding to the shISCU2-directed shRNA were introduced into the wild-type and mutant *ISCU2* open reading frame to obtain plasmids encoding RNAi-resistant ISCU2 versions.

**Cell culture and transfection.** Human cervix carcinoma cells (HeLa cells: European Collection of Authenticated Cell Cultures (ECACC), cat. #93021013) were cultured and transfected as described previously[43,55]. In short, $5 \times 10^6$ cells were resuspended in transfection buffer (21 mM Hepes, 137 mM NaCl, 5 mM KCl, 0.7 mM Na$_2$HPO$_4$, and 6 mM glucose), supplemented with 10–30 µg of each of the required plasmids, and transfected by electroporation (using an EASYJect device). Three days later, a second round of transfection was carried out to intensify ISCU1/2 depletion by another 3 days of growth.

**Harvesting and fractionation of HeLa cells.** HeLa cells were harvested by trypsination, washed with phosphate-buffered saline, and samples of dry cell pellets were stored frozen until preparation for immunoblotting. For crude cell fractionation, freshly harvested cells were suspended in fractionation buffer (25 mM Tris-HCl, pH 7.4, 250 mM sucrose, 1.5 mM MgCl2, 0.01 % digitonin, 1 mM PMSF), soluble cytosolic constituents were separated from the membrane/organelle-containing fraction by centrifugation at $15,000 \times g$ in a refrigerated table-top centrifuge[56]. Submitochondrial protein localization was performed with 15 µL of freshly prepared organellar digitonin fractions at a total protein concentration of 5 µg/µL[57]. For mitochondrial swelling and rupture of the outer mitochondrial membrane samples were mixed with 325 µL of 5 mM HEPES (pH 7.4), whereas total membrane solubilization was achieved by addition of Triton X-100 at a final concentration (fc) of 0.2%. Target protein accessibility was tested by addition of proteinase K (fc 66 µg/mL) and incubation at 30 °C for 30 min. Subsequently, samples were precipitated by trichloroacetic acid (TCA, fc 10%), washed with ice-cold acetone, and solved in Lämmli buffer for immunoblotting.

**Determination of enzyme activities.** Aconitase (Aco) activity was determined by a coupled aconitase-isocitrate dehydrogenase assay including 1.25 mM NADPH, 300 µM cis-aconitate, 200 mU isocitrate dehydrogenase (IDH) in a buffer consisting of 100 mM triethanolamine (pH 8.0), 1.5 mM MgCl2, and 0.1% Triton X-100[56]. Mixtures lacking cis-aconitate and IDH were used to determine aconitase-independent NADPH oxidation ($\lambda = 340$ nm, $\varepsilon = 6.22$ mM$^{-1} *$ cm$^{-1}$). Succinate dehydrogenase (SDH) activity was measured in an assay mixture consisting of 50 mM Tris/SO4 (pH 7.4), 0.1 mM ethylenediaminetetraacetate (EDTA), 70 µM dichlorophenol-indophenol (DCPIP), 50 µM decylubiquinone, 0.2% sodium succinate, and 0.1% Triton X-100[57,58]. SDH-independent DCPIP reduction ($\lambda = 600$ nm, $\varepsilon = 21$ mM$^{-1} *$ cm$^{-1}$) was monitored by addition of 0.2% sodium malonate. Cytochrome-c oxidase (COX) activity was determined in the organellar cell fraction by measuring the oxidation of reduced cytochrome *c* (about 1 mg/mL) in 50 mM 2-(N-morpholino)ethanesulfonic acid (MES, pH 6.6), 50 mM sodium chloride (NaCl), 0.5 mM dodecylmaltoside, and 1% bovine serum albumin[59]. Oxidation of cytochrome-c by air oxygen was examined in the absence of cell material ($\lambda = 550$ nm, $\varepsilon = 19.1$ mM$^{-1} *$ cm$^{-1}$). Citrate synthase (CS) activity was assessed in a reaction mixture composed of 50 mM Tris/HCl (pH 8.0), 100 mM NaCl, 0.5 mM 5,5′-Dithio-bis-(2-Nitrobenzoic acid) (DTNB), 125 µg/mL acetyl-CoA, 200 µg/mL oxaloacetic acid, and 0.1% Triton X-100[60]. CS-independent decomposition of DTNB was analyzed in the absence of oxaloacetic acid ($\lambda = 412$ nm, $\varepsilon = 13.3$ mM$^{-1} *$ cm$^{-1}$). The reaction mixture for the determination of lactate dehydrogenase (LDH) activity consisted of 50 mM Tris/HCl (pH 7.4), 1 mM EDTA, 750 µM sodium pyruvate, and 150 µg/mL NADH. A potential NADH oxidation by air oxygen was taken into account by measuring the absorption in the absence of cell material ($\lambda = 340$ nm, $\varepsilon = 6.22$ mM$^{-1} *$ cm 1).

**Expression and purification of recombinant proteins.** Proteins for structure determination were purified similarly to previously published methods[28]. The mature form of mitochondrial ISCU2 was used to reconstitute the (NIAU2)$_2$ complex. For expression of ISCU2, plasmid p24am (gift from Dr. Kuanyu Li, Nanjing University China) encoding ISCU2 residues 35-168 with a start Met and a C-terminal His$_6$-tag was used. Purification of ISCU2 mutant proteins M140I, L35, L35H36, and Y35D was carried out using the same protocol. Proteins were concentrated immediately after purification with slow buffer exchange in the concentrator (3 kDa cut-off, Millipore) to remove excess of imidazole. Final protein concentration (16–18 mg/ml) was measured using OD$_{280}$ absorbance. Proteins for biochemical analyses were expressed in *E.coli* BL21 DE3 cells by growth in Terrific Broth supplemented with the respective antibiotics (Supplementary Table 4). Expression was induced by the addition of 1 mM IPTG at an OD$_{600}$ ~0.8–1 and continued overnight at 28 °C. Cells were pelleted by centrifugation and stored at −20 °C until purification.

For purification cells were thawed at room temperature, resuspended in 35 mM Tris-HCl pH 8 containing 300 mM NaCl, 5% (w/v) glycerol and 10 mM imidazole (buffer P) and lysed by sonication (SONOPULS mini20, BANDELIN electronic GmbH & Co. KG, Berlin, Germany). Cell debris was removed by centrifugation at 40,000 × $g$ for 45 min. For all proteins except for FDX2 the supernatant containing soluble protein was subjected to Ni-NTA affinity chromatography (Prepacked HIS-Trap 5 ml ff crude; GE Healthcare) and subsequent size-exclusion chromatography (SEC) (16/60 Superdex 75 or 200; GE Healthcare) on an Äkta Purifier 10 or Äkta Pure systems (GE Healthcare). For SEC and storage, the buffer was adjusted to 35 mM Tris-HCl pH 8, 150 mM NaCl and 5% (w/v) glycerol (buffer S). Elution from the Ni-NTA matrix was achieved by applying a linear gradient from 0.01 to 1 M imidazole. Proteins typically eluted between 120 and 250 mM of imidazole.

Exceptions from the standard procedure: Human NFS1-ISD11-ACP were co-expressed (NFS1 and ISD11 were inserted into pET-Duet MCSI and MCSII, respectively, and ACP into pRSF-Duet MCSI, for detailed sequence information refer to Supplementary Table 4) and co-purified using the His$_6$-tag fused to ISD11 in buffer P additionally containing 5 mM PLP. Subsequently, the complex was purified to homogeneity by SEC in buffer S, and elution fractions contained a bright yellow protein. ISCU2 and variants were treated with 5 mM EDTA, 5 mM KCN and 5 mM DTT prior to SEC to remove potentially bound metal ions and/or polysulfides. FXN was treated with self-made recombinant TEV protease, β-mercaptoethanol and DTT prior to SEC to cleave the N-terminal His$_6$-tag (which otherwise renders the protein inactive) and to remove potential metal ion contaminations. FDX2 was purified using anion exchange chromatography (AEC) and subsequent SEC. For AEC, 35 mM Tris-HCl pH 8 containing 20 mM NaCl (buffer A) was used. Elution was performed applying a linear gradient increasing the NaCl concentration from 0.02 to 1.0 M. FDX2 typically eluted at a NaCl concentration of 150–300 mM. SEC was performed as outlined above and yielded a dark brown protein solution.

**Protein structure determination.** All the (NIAU2)$_2$ complexes containing either wild-type ISCU2 or its variants were crystallized using the same conditions. Concentrated complex solution (16–18 mg ml$^{-1}$) was dispensed into a 96-well sitting drop plate using a Gryphon crystallization robot (ArtRobbins Instruments, Sunnyvale, CA). Each drop contained 300 nL of protein solution and 300 nL of mother liquor. The initial crystals appeared at 15 °C in wells containing 0.1 MES pH 6.0 and 22% PEG-400 in PEG-Rx screen (Hampton Research, Alison Viejo, CA). The best diffracting crystals were grown in optimized condition of 0.1 M MES pH 6.5, and 22.5% PEG-400 at 15 °C. Crystals were harvested at 4 °C and stored in liquid nitrogen with 20% glycerol as cryoprotectant. Crystals diffracted to 1.57–1.95 Å in P4$_1$2$_1$2 space group, with cell dimensions similar for all wild-type and mutant versions of (NIAU2)$_2$ complexes $a$ = ~86.4 Å, $b$ = ~86.4 Å, $c$ = ~245.8 Å.

Diffraction data were collected using LRL-CAT 31-ID beamline at the Advanced Photon Source (Argonne National Laboratory, IL) for (NIAU2)$_2$ with wild-type ISCU2 using Rayonix MX225 CCD detector. Diffraction data for crystals of the (NIAU2)$_2$ complex containing ISCU2 mutant proteins were collected using 08ID beamline at the Canadian Light Source, CMCF sector, Saskatoon, SK, using Pilatus3 S 6 M detector. Data were integrated and scaled using the XDS package[61]. Initial phases were obtained by using the previously published model of human (NIAU1)$_2$ complex (PDB ID: 5WKP) for molecular replacement and Phaser MR package[62] (Phaser MR within ccp4i package v 7.1). The manual rebuilding of the model was performed using COOT[63] (v 0.9.5) and structures were refined using the Phenix package[64] (v 1.19).

**Crystallographic data and protein structure deposition.** The final models of human (NIAU2)$_2$ complexes contain almost all amino acid residues within the protein chains. However, the large flexibility of the NFS1 cysteine loop led to its distortion in the models containing wild-type, ISCU2-M140I, and ISCU2-Y35D. The cysteine loop is only partially visible with Cys381 pointing toward PLP in the NFS1 active site. The residues T382SASLE387 could not be modelled due to poor electron density. In the models for human (NIAU2)$_2$ complexes with ISCU2-L35 and ISCU2-L35H36 mutant proteins, the NFS1 cysteine loop is entirely visible pointing toward the active site of ISCU2. Cys381$^{NFS1}$ contacts Lys135$^{ISCU2}$ without metal present in the ISCU2 active site. Data collection and refinement statistics are

provided in Supplementary Table 2. All models were deposited in the PDB data base with PDB IDs as follows: (NIAU2)$_2$ with wild-type ISCU2: 6W1D; (NIAU2)$_2$ with ISCU2-M140I: 6UXE; (NIAU2)$_2$ with ISCU2-L35: 6WI2; (NIAU2)$_2$ with ISCU2-L35H36: 6WIH; (NIAU2)$_2$ with ISCU2-Y35D: 7RTK.

**Spectroscopic methods.** UV/VIS absorption and CD spectroscopy was performed anaerobically at room temperature in a quartz cuvette with 10 mm (UV/VIS) or 5 mm (CD) path length sealed with a rubber capping on a Jasco V550 and Jasco J-815 (JASCO Deutschland GmbH, Pfungstadt, Germany), respectively.

**Persulfide transfer assay.** All persulfide transfer assays[35] were done anaerobically at room temperature using degassed buffer T (50 mM Tris-HCl pH 7.4, 150 mM NaCl, 200 μM Na-ascorbate, 200 μM FeCl$_2$) and a final reaction volume of 40 μL. The indicated proteins were added (40 μM ISCU2 or the respective variant; 80 μM FXN and 80 μM (NIA)$_2$ or the indicated variant). After incubation for at least 5 min, sulfur transfer was initiated by adding 2 mM L-cysteine, and transfer was quenched after 10 s using 4.44 mM EZ-Link maleimide-polyethylene-glycol$_{11}$-biotin (MPB, Thermo Scientific), i.e., 1.5 equivalents of total thiol concentration. As a negative control, either no (NIA)$_2$ or no L-cysteine was added. After 1 min, SDS was added to a final concentration of 2% (w/v), and after another 10 min of incubation, samples were removed from the anaerobic chamber. Reaction aliquots containing 2 μg ISCU2 were incubated in sample buffer with 50 mM TCEP for 15 min at room temperature. Analysis by SDS-PAGE (16% polyacrylamide tricine gel) and staining using InstantBlue (Expedeon, #ISB1L) followed manufacturer's instruction. Samples for persulfide reduction by FDX2 were treated the same way except, 10 s after cysteine addition, 200 μM NADPH, 1 mM MgCl$_2$, 0.4 μM FDXR and 20 μM FDX2 were added simultaneously. The solution was incubated for 3 min before quenching with MPB11 and SDS as described before.

**NFS1 desulfurase activity.** Cysteine desulfurase activity of purified (NIA)$_2$ complexes and NFS1 variants was determined by DTT-dependent sulfide generation as published[31,65] with minor modifications. Purified protein was incubated at 30 °C in 25 mM Tricine, pH 8.0, 1 mM DTT, and 1 mM L-cysteine. After 20 min the reaction was stopped by addition of 4 mM N,N-dimethyl-p-phenylenediamine sulfate (in 7.2 N HCl) and 3 mM FeCl$_3$ (in 1.2 N HCl). Samples were incubated 20 min in the dark and the amount of methylene blue was determined spectrophotometrically at 670 nm.

**Affinity measurements.** Protein-protein interactions were measured using microscale thermophoresis on a Monolith 1.15 (Nanotemper Technologies, Munich, Germany). Measurement settings: LED power at 30%, laser power at 75%, temperature fixed at 21 °C. (NIA)$_2$ and variants were labelled according to manufacturer's instructions (Monolith Protein Labeling Kit RED-NHS 2nd Generation, MO-L011, Nanotemper Technologies). 200 nM labelled protein was titrated with a 1:1 dilution series of wild-type ISCU2 or the respective variant starting from 100 μM to 200 μM as indicated. Data were analyzed using Origin 8 G (OriginLab Corporation, Northampton, MA, USA).

**Chemical and enzymatic Fe/S cluster reconstitution on ISCU2 proteins.** Chemical reconstitution was carried out under anaerobic conditions as described previously[66]. Proteins were pre-incubated with 5 mM DTT for 1 h on ice. Five equivalents of ferric ammonium citrate were added, and after 15 min at room temperature 5 equivalents of Li$_2$S were added to the sample in three steps with mixing in between. After 15 min of reconstitution samples were desalted using a PD-10 column.

Prior to the enzymatic reconstitution, ISCU2 proteins were treated with DTT overnight inside the anaerobic chamber at 11 °C, and were subsequently anaerobically gel-filtrated on an UltiMate 3000 UHPLC system (Thermo Fisher Diagnostics GmbH, Germany) using a 16/60 Superdex 75 SEC column (GE Healthcare) (for a representative result see Source Data). Enzymatic reconstitution of [2Fe-2S] cluster by the core ISC complex was followed by CD spectroscopy as described[25,28,66] with the exception that GSH and MmGrx1 (mouse glutaredoxin 1) were added to the samples. A usual setup contained: 0.8 mM ascorbate, 0.3 mM FeCl$_2$, 0.5 mM NADPH, 0.2 mM MgCl$_2$, 5 mM GSH, 0.5 μM FDXR, 5 μM FXN, 5 μM FDX2, 5 μM MmGrx1, 5 μM (NIA)$_2$ (with wild-type or mutant NFS1), 75–100 μM ISCU2 (wild-type or mutant). Enzymatic reconstitutions analyzed by aSEC were carried out identically, yet no GSH and MmGrx1 were added.

For determination of the origin of ISCU2 within the dimeric holo-product (Fig. 6b,c; Supplementary Fig. 10) purified (NIA)$_2$, ISCU2-HIS, FXN, and FDX2 (75 μM each) were mixed anaerobically with 0.8 mM ascorbate, 0.3 mM FeCl$_2$, 0.5 mM NADPH, 0.2 mM MgCl2, 5 mM GSH, 0.5 μM FDXR in buffer S and incubated for 5 min. Subsequently, 75 μM free cysteine and 150 μM non-tagged ISCU2 were added simultaneously. After incubation for 10 min at room temperature the mixture was subjected to anaerobic aSEC (see next chapter). The initial mixture as well as the three prominent peaks (corresponding to (NIA)$_2$ complex, ISCU2 dimer and monomer) were analyzed by SDS-PAGE and subsequent Coomassie staining (using InstantBlue; Expedeon, #ISB1L) or Western blot analysis using an anti-HIS tag antibody.

**Analytical size-exclusion chromatography**. Enzymatically reconstituted ISCU2 and variants were subjected to analytical size-exclusion chromatography (aSEC) inside an anaerobic chamber (Coy Laboratory products, Inc., Grass Lake, MI, USA) on an UltiMate 3000 UHPLC system (Thermo Fisher Diagnostics GmbH, Germany) using a 10/300 Superdex 75 aSEC column (GE Healthcare) equilibrated with buffer S and calibrated using the LMW kit (28-4038-41, GE Healthcare). Wavelengths recorded: 280 nm (protein), 320 nm and 420 nm (Fe/S cluster).

**Di-tyrosine formation by UV irradiation**. 25 μM ISCU2 proteins in buffer S were chemically reconstituted, and di-tyrosine formation was induced by UV irradiation by a transilluminator (Vilber Lourmat FLX 20 L) for 5 min at 5.000 J/cm$^2$ on ice. Fluorescence spectra were recorded on a Jasco FP 6300 spectrofluorometer (settings: $\lambda_{Ex}$: 320 nm, bandwidth 5 nm; $\lambda_{Em}$: 380-480 nm), and normalized to zero at 380 nm.

**Antibodies**. The primary antibodies listed in Supplementary Table 5 were used for immunoblotting. Peroxidase-conjugated goat, anti-rabbit (#170-6515) and anti-mouse (HRP #170-6516) antibodies (Biorad, Germany) were used as secondary reagents.

**Reporting summary**. Further information on research design is available in the Nature Research Reporting Summary linked to this article.

## Data availability

The crystallographic datasets generated and analyzed during the current study are available in the Protein Data Bank (PDB, www.rcsb.org). PDB IDs are as follows: (NIAU2)$_2$ with wild-type ISCU2: 6W1D; (NIAU2)$_2$ with ISCU2-M140I: 6UXE; (NIAU2)$_2$ with ISCU2-L35: 6WI2; (NIAU2)$_2$ with ISCU2-L35H36: 6WIH; (NIAU2)$_2$ with ISCU2-Y35D: 7RTK; (NIAU1)$_2$: 5WKP; (NIAU2)$_2$ plus bound frataxin: 6NZU. All other datasets generated and analyzed during the current study are either shown in the source data file or are available from the corresponding author on reasonable request. Source data are provided with this paper.

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

## Acknowledgements

We thank M. Stümpfig and N. Richter for technical support. R.L. acknowledges generous financial support from Deutsche Forschungsgemeinschaft (SFB 987, SPP 1710, and SPP 1927), and from the Behring-Röntgen Foundation, and the LOEWE program of state Hessen. We acknowledge networking support from the COST Action FeSBioNet (Contract CA15133). M.C. acknowledges financial support from CIHR (grant MOP-48370). Data collection was performed at the beamlines: 31-ID, LRL-CAT, Advanced Photon Source, Argonne, IL, USA, and 08ID-1, Canadian Light Source, which is supported by the Canada Foundation for Innovation, Natural Sciences and Engineering Research Council of Canada, the University of Saskatchewan, the Government of Saskatchewan, Western Economic Diversification Canada, the National Research Council Canada, and the Canadian Institutes of Health Research. We acknowledge the Protein Characterization and Crystallization Facility, College of Medicine, University of Saskatchewan for access to the crystallization robot and multi-angle light scattering instrumentation and support.

## Author contributions

S.A.F., V.S., and N.K. performed biochemical experiments. S.A.F., V.S., and R.L. designed and analyzed biochemical experiments. M.T.B. crystallized the proteins. M.T.B. and M.C. analyzed the crystallographic data. C.S. and O.S. performed and, together with R.L., analyzed human cell culture experiments. D.R.W. contributed to experimental design. N.K. and U.M. performed and, together with R.L., analyzed yeast cell culture experiments. R.L. and M.C. acquired funding. S.A.F., M.T.B., C.S., O.S., M.C., and R.L. wrote the paper. All authors read and commented on the paper.

## Funding

## Competing interests

The authors declare no competing interests.
