## [Peer Review File · Nature Communications]

REVIEWER COMMENTS

Reviewer #1 (Remarks to the Author):

In this manuscript, Dr Svein-A Freibert and co-workers report on the role of the conserved tyrosine residue 35 from ISCU2 in stabilizing dimerization hence triggering formation of [2Fe-2S] clusters in eukaryotes. The importance of this residue was already reported by Tanaka and coworkers in 2019 (reference 40 in the present manuscript). However, its precise role was not fully understood. Therefore, the authors first confirmed the key role of residue Y35 using in vivo RNAi depletion-complementation. They subsequently determined the crystal structure of the (NIAU2)₂ complex and of several variants and based on the observed electron density, they concluded that the Y35 containing N-terminal stretch is flexible, at odds with what was previously reported when using cryo-EM technique. Subsequently, they investigated the role of the NFS1 residue W39, which interacts with Y35 and demonstrated that this residue is not key to the FeS cluster assembly process. Then, using in vitro assays, the authors showed that residue Y35 does not play any role in the early steps of the reaction, supporting its involvement during the ISCU2 dimerization step. Site-directed mutagenesis to either impair the reaction or restore it through the installation of a salt-bridge, combined to UV-induced crosslinking showed that indeed, Y35 is important to ISCU2 dimerization leading to formation of the [2Fe-2S] cluster. This conclusion is an important milestone in our understanding of the FeS cluster assembly process. However, beside not being novel, in its present form, the manuscript raises questions that need to be addressed before any publication. Overall, the reviewer suggests this work would be published in a more specialized journal instead of Nature Communications.

Here are the main points the reviewer would raise to the authors' attention:

1- The present work demonstrates the key role of Y35 in inducing ISCU2 dimerization. It was already known that such dimerization induces [2Fe-2S] cluster formation. However, the authors do not provide any mechanistic information about how this Y35 residue induces the needed dimerization. What does trigger the dimerization? Is it a dynamic mechanism leading to ISCU2 dimers often empty and sometimes pre-loaded with Fe and S? Conversely, does ISCU2 need to bind Fe and S prior to dimerization? Is Y35 the trigger or a stabilizer?

2- It is not really clear to the reviewer understanding why a M140I variant structure is reported in the extended data Table 2. Indeed, this structure is not discussed in the main text and does not seem to be related to the present study. Please remove that crystal structure.

3- Page 6, the authors compare the new NIAU2 crystal structure with the NIAU1 one. It is not clear why there is no comparison with the NIAU2 one the authors determined using cryo-EM. Furthermore, there is a clear lack of thorough structural analysis when reporting the 6.4° rotation of the ISCU2 position when compared to the ISCU1 position. Indeed, in addition to missing to mention

whether such rotation was already observed in the cryo-EM structure, nothing is described in the manuscript to take into account variations that could be induced by differences in crystal packing.

4- The main conclusion from the crystallography analysis seems that the Y35 residue is disordered in the crystal and may adopt two favored conformations. This is not really clear from the electron density map presented in Fig 2c. Stereo-views might help better figure out the validity of the proposed model. However, there are apparently few blobs and no details are provided by the author to support their current model. About Fig 2c, the comparison of the three variants is even more difficult to assess because the panels do not correspond to the same orientation. In addition, it is really often that N- and/or C-terminal stretches are disordered (as already observed for ISCU1). Therefore, it is really difficult to conclude anything from the presented structural data.

5- The rationale supporting the different variants YHKK versus LSTQ or LHTQ is not really clear. What the authors really expected from these variants? In the present version of the manuscript, the reviewer feels that the authors performed different variants to mimic the ISCU1 protein (despite the orientation of the ISCU protein toward the NSF1 core was different) and that in the end they try to fit the corresponding structure into another story, yet not really convincingly. Indeed, as stated by the authors, the only relevant conclusion from this structural part is: “the high resolution 3D architecture of the (NIAU2)₂ complexes shows a flexible N-terminal region, yet otherwise is highly similar to the (NIAU1)₂ structure.” The rest of the discussion sounds highly speculative and poorly supported by any data. This feeling is further supported by the M140I variant that has nothing to do with the present study.

6- Regarding the role of W97, the reviewer disagrees with the authors' conclusions. Extended Fig 6b indicates that the activity of the W97K variant is not affected, while that of the W97D is. The authors seem to neglect the possibility that lysine may establish a cation- π interaction, while aspartate cannot. At least a crystal structure of the W->K variant would be expected to further investigate that possibility and see whether this stabilizes Y35 in one orientation.

7- In the discussion section, the authors claim that: “Our new findings together with numerous published studies (reviewed in 4) allow us to propose a molecular mechanism for the complete cycle of mitochondrial [2Fe-2S] cluster biogenesis”. In fact, the current contribution is limited to the identification of the role of Y35 as a key residue to stabilize ISCU2 dimerization. All the other parts of the proposed mechanism were already known. Strikingly, the authors never mention, in the text, when and how iron comes in ISCU. They only mention it in Fig 6b. There isn't any mention of a possible role of iron binding and persulfide transfer and reduction in the signaling for triggering dimerization. Did the authors test for spontaneous ISCU2 dimerization in the absence of L-cysteine and/or iron?

Considering all these remarks, the reviewer does not support publication of the manuscript in its current form.

Reviewer #2 (Remarks to the Author):

Freibert et al. describe the characterization of a variant of IscU that is defective in participating in the initial steps of Fe-S cluster assembly in the mitochondria. This contribution from the Lill's group builds on their previous work and advances our understanding of how Fe-S clusters are formed in eukaryotic systems. Overall, the findings described in this study are in line with earlier reports in the literature that identified the conserved tyrosine residue at the N-terminal of IscU and reported the formation of a 2Fe-2S cluster on an IscU dimer. The experimental data and presentation are well-sounded to support this major finding but do not explain all the aspects of the detailed model put forward by the authors. The manuscript narrative, however, makes some additional general statements that go beyond the scope of this study and, in the end, weakens the overall quality of this report. While merits publication, this contribution is built on previous findings reported and proposed by others and attempts to sell a model for a process that has not been completely elucidated. My overall recommendation is to revise the document to address these concerns and provide a discussion narrative that is more conservative and focused on the results provided in this study.

Main points

1- While the biochemical data provided in this study is very compelling, the authors have not explored how this variant may affect the involvement of frataxin and ferredoxin in earlier stages of this process. Experiments described in Fig.3AB and 4AB were completed in the absence of iron, which is known to affect the kinetic profile of these reactions (ref 32, 36). Another aspect that was not explored was the potential involvement of this residue in the reduction step promoted by ferredoxin. Thus, structural and mechanistic information is limited to investigate sulfur transfer and cluster accumulation and not the synthesis of the cluster itself. No experimental data is provided to show that IscU2 monomer, while in a complex, interacts with another monomer of IscU from a secondary core complex (Fig.6 step 3).

2- While dimerization of IscU monomers, each coordinating 1S and 1Fe has been proposed earlier (ref 36), the data shown in this study provides a good contribution as it defines the role of Y35 as a structure element enabling this interaction. However, some of the experimental design (when indicated) to probe these reactions are performed under sub-stoichiometric amounts of IscU2 (40 μ M IscU2 vs 80 μ M of the NIA). Since free IscU2 is a product of this biosynthetic cycle (Fig6 step 2), one is left with a question about the effect of this reaction if an excess of IscU2 increases the rate of this reaction. How can the authors rule out a model that involves a free form of IscU (apo or Fe-bound) from interacting with the core complex and leading to the formation of an IscU dimer?

3- The authors have not provided a comparative study to justify general comparisons with various bacterial systems that likely employ distinct mechanistic strategies to assemble clusters. While attempting to generalize this process, the authors create a problem for themselves since additional literature available for other systems not necessarily fits their proposed model. Therefore, I strongly recommend the authors to limit the discussion to the system explored in this study and minimize speculation on whether these findings are applicable to other systems.

3.1 As noted, Tyr35 is conserved in other species containing IscU and IscU like sequences. As previously reported (ref 40), this equivalent residue in bacteria also affects function. However, suppressor mutants were identified containing substitutions within IscS A349. This evidence points to the potentially distinctive role for this residue in bacterial IscU and perhaps mechanistic differences in prokaryotic systems.

3.2 Characterization of prokaryotic IscU proteins has demonstrated the ability of monomeric forms of these proteins to coordinate Fe-S clusters (from in vivo and in vitro forms). These proteins can then form a 4Fe-4S cluster in reactions involving ferredoxin. Relevant to the Y35 role, recent work by Takahashi and co-workers reported on the structure of a dimer of IscU coordinating 2 [2Fe2S] clusters (<https://doi.org/10.1021/acs.biochem.1c00112>). Collectively these reports suggest that bacterial ISC systems, which do not have additional Fe-S cluster biosynthetic components like the mitochondria ISC, can coordinate a 2Fe-2S per monomer and promote the synthesis of a 4Fe-4S cluster.

3.3 The discussion about the potential role of SufU in functioning as a scaffold disregards recent literature describing genetic, biochemical, and structural differences between bacterial IscU and SufU from a number of species. The location of equivalent Y in the SufS-SufU structure is likewise remote from the site of sulfur transfer. While it may affect interactions with SufS, it remains to be determined if substitution of this residue will affect zinc binding and sulfur transfer reactions to downstream scaffold SufBCD, which is absent for the species investigated in this study.

3.4 The structure of the archaea *A. fulgidus* IscS-IscU complex shows the coordination of an Fe-S cluster to a monomer of IscU in which a cysteine residue of IscS provides the fourth ligand. The authors are correct to indicate the IscS is inactive as it lacks the conserved lysine coordinating the PLP(PMP) cofactor. In this particular case, it has been proposed that the activity of a cysteine desulfurase may be dispensable due to the organism's lifestyle. Regardless, this structural model provides a valid example that an IscU monomer is capable of coordinating a cluster. And again, this structure highlights mechanistic differences from eukaryotic and prokaryotic systems and perhaps suggests their functions have evolved to meet distinct biosynthetic requirements and environmental factors.

Overall, a suggested approach for revising the discussion is to acknowledge that these systems likely evolve to perform distinct functions. The case of *Mycoplasma* noted in the manuscript is a good example of that. Prokaryotic Fe-S assembly is not confined to organelles and employs considerably fewer components. Therefore these components may accumulate multiple and distinct functions. Presenting a narrative that invalidates or discredits the findings reported for other systems without providing experimental evidence does not improve the impact of this publication or inspire others to define unresolved aspects of these complex pathways.

Minor points

- Figure 3C legend or figure needs to indicate that frataxin was added.

- Methods describing cluster enzymatic reactions do not include a description of reaction ingredients and concentrations. Please revise.

- Page 10 second paragraph discussion – new high resolution structures (as low as 1.57 Å)

Response to Reviewers' comments Manuscript NCOMMS-21-14172, Freibert et al.

We thank the Reviewers for their careful reading, the overall positive evaluation and the helpful criticism of our new work. We have addressed and clarified all mentioned issues in our revised version, and have used the helpful suggestions and constructive concerns to now provide additional experimental data (both structural and biochemical) and better explanations. All new data fully support our mechanistic conclusions and the proposed model. We hope that, after substantial revision and further polishing of the text, our manuscript will now be acceptable for publication in the *Nature Communications*.

Note: Reviewers' original text is in italics; our response is in plain text. The changes in our manuscript during revision are highlighted in file Freibert.revision-changes.pdf to ease the reviewing process.

REVIEWER COMMENTS

Reviewer #1 (Remarks to the Author; see attached PDF for formatted version):

In this manuscript, Dr Svein-A Freibert and co-workers report on the role of the conserved tyrosine residue 35 from ISCU2 in stabilizing dimerization hence triggering formation of [2Fe-2S] clusters in eukaryotes. The importance of this residue was already reported by Tanaka and coworkers in 2019 (reference 40 in the present manuscript). However, its precise role was not fully understood. Therefore, the authors first confirmed the key role of residue Y35 using in vivo RNAi depletion-complementation. They subsequently determined the crystal structure of the (NIAU2)₂ complex and of several variants and based on the observed electron density, they concluded that the Y35 containing N-terminal stretch is flexible, at odds with what was previously reported when using cryo-EM technique. Subsequently, they investigated the role of the NFS1 residue W39, which interacts with Y35 and demonstrated that this residue is not key to the FeS cluster assembly process. Then, using in vitro assays, the authors showed that residue Y35 does not play any role in the early steps of the reaction, supporting its involvement during the ISCU2 dimerization step. Site-directed mutagenesis to either impair the reaction or restore it through the installation of a salt-bridge, combined to UV-induced crosslinking showed that indeed, Y35 is important to ISCU2 dimerization leading to formation of the [2Fe-2S] cluster. This conclusion is an important milestone in our understanding of the FeS cluster assembly process. However, beside not being novel, in its present form, the manuscript raises questions that need to be addressed before any publication. Overall, the reviewer suggests this work would be published in a more specialized journal instead of Nature Communications.

We thank the Reviewer for the judgement of our study as “an important milestone in our understanding of the FeS cluster assembly process”. We therefore are confident that with our additional data and explanations, the Reviewer will be able to support publication of our findings. We agree that the physiological importance of the N-terminus and its Tyr residue has been described earlier by us for yeast Isu1/2 (Gerber 2004) and by Takahashi in 2019 for *E. coli* IscU, but since the biochemical role of the N-terminal Tyr was completely unknown and on the basis of the current discussion in the FeS field that bacterial Fe/S cluster formation might deviate from the mitochondrial pathway (see Reviewer 2), we think this novel information for the human Tyr35^{ISCU2} is indeed important and crucial. Our new biochemical findings assign a defined mechanistic (and previously unknown) function to Tyr35. Moreover,

our data (indirectly) provides interesting information concerning the functionality of the human cytosolic ISCU1 lacking this residue.

We do not think that there is a major disagreement in the structural information from the cryo-EM or X-ray data, with the important difference that our new X-ray structures have been determined up to 1.6 Å resolution. At this resolution, the majority of the residues are very well resolved in the electron density and poor density clearly indicates partial disorder. In all the datasets we have collected, the density for the first two residues of ISCU2, i.e. Tyr 35-His36, is weak and indicates two major conformations of Tyr35. The electron density for all subsequent residues is extremely well defined. In contrast to the Reviewer's impression, the discrepancy of our X-ray structure with the cryo-EM-derived model is not supported by the cryo-EM map (see new Supplementary Fig. 4). There is clear additional density in the cryo-EM map indicating an alternative conformation of Tyr35. Since this may possibly be less clear than in the high resolution X-ray data sets, this alternative orientation was not incorporated into the cryo-EM model. We have addressed this issue in Results.

Here are the main points the reviewer would raise to the authors' attention:

1- The present work demonstrates the key role of Y35 in inducing ISCU2 dimerization. It was already known that such dimerization induces [2Fe-2S] cluster formation.

From early pioneering work of Dean and Johnson (and also Vickery), it was indeed known that bacterial dimeric IscU binds a stable [2Fe-2S] cluster upon chemical reconstitution (dimer inferred by Fe and S stoichiometries and protein-FeS extinction ratios; now cited: Agar JACS 2000, Chandramouli Biochem. 2007). It was not known, however, that cluster binding 'induces' dimerization because gel filtration experiments were not performed in these early studies. The dimerization of mitochondrial IscU-like proteins has been suggested in our paper by Webert et al. 2014 in Nature Commun. Later, Gervason et al. (Nature Commun 2019) also proposed dimer formation based on indirect findings (without referring to our paper). In our 2014 paper, we indirectly deduced holo-ISCU2 dimer formation with a bound [2Fe-2S] cluster from Fe and S stoichiometries, FeS cluster saturation, and crosslinking studies. We now directly (and thus much more convincingly) follow the process of ISCU2 dimerization during the various steps of [2Fe-2S] cluster formation by using gel filtration (SEC) in which we can assign monomer and dimer and the bound cluster by multi-wavelength spectroscopy. In the new manuscript we studied for the first time the mechanism of how dimerization is triggered by Tyr-Tyr interaction, and how [2Fe-2S] cluster formation (note that cluster is only seen on the ISCU2 dimer, not on the monomer) is mechanistically achieved. This data adds new key aspects to the mechanistic understanding of core ISC function in [2Fe-2S] cluster assembly, an essential reaction in Biology.

However, the authors do not provide any mechanistic information about how this Y35 residue induces the needed dimerization. What does trigger the dimerization? Is it a dynamic mechanism leading to ISCU2 dimers often empty and sometimes pre-loaded with Fe and S? Conversely, does ISCU2 need to bind Fe and S prior to dimerization? Is Y35 the trigger or a stabilizer?

These questions are well-taken, and our newly added data clarify these issues. Using tagged and untagged ISCU2, we show in the new Fig. 6b and 6c and new Supplementary Fig. 10 that only ISCU2-His that is already bound to the (NIA)₂ complex can give rise to the holo-ISCU2 dimer with the bound [2Fe-2S] cluster. Free apo-ISCU2 (no tag) added exogenously to the (NIAU2)₂ complex does not enter the dimer, yet can associate with the freed binding sites on the (NIA)₂ complex once pre-bound ISCU2-His has left for dimerization. In our view this experimental data very strongly supports our mechanistic model proposed in Fig. 7.

Previous work from our and other labs (Barondeau, D'Autreaux, Pain) has shown that (NIA)₂-prebound ISCU2 holds both sulfide and iron (already mentioned in Abstract). As the experiment in Fig. 6b,c demonstrates, this complex is the substrate for the fusion of two Fe-loaded ISCU2 attached to S-loaded (NIA)₂, a reaction facilitated by interaction between two Tyr35. Since Tyr35 is absolutely essential for this step (no dimer formation with ISCU2-Y35D/K variants; Fig. 5d), this residue functions as a 'trigger' to induce rapid dimerization (see the title of our manuscript). We thank the reviewer for suggesting improved explanation of this mechanistic issue (in Results and Discussion). The 'trigger' interpretation is further supported by *in vitro* chemical reconstitution which is able to produce holo-ISCU2-Y35A dimer (Supplementary Fig. 8b), yet in a non-physiological, DTT-dependent, and rather slow reaction. While being artificial, the latter experiment shows that the mutated protein in principle is well-folded and can hold a cluster, but is unable to receive its cluster via the physiological (NIA)₂-catalyzed pathway.

2- It is not really clear to the reviewer understanding why a M140I variant structure is reported in the extended data Table 2. Indeed, this structure is not discussed in the main text and does not seem to be related to the present study. Please remove that crystal structure.
We agree (and apologize) that this mutant was not described in detail in our manuscript (this shortcoming was mainly due to space limitations). The structure of this mutant complex was determined in a search for the clues making this ISCU2 variant independent of frataxin. We found that the mutation does not affect the global or local structure of the complex, and hence does not provide a simple explanation for the observed functional change. We now have mentioned this issue in the text. We included the M140I structure because this complex was determined at the highest resolution, and hence is interesting structurally by itself. We think this justifies its inclusion in the manuscript, even though the Reviewer is right that it does not closely relate to the main topic of our story.

3- Page 6, the authors compare the new (NIAU)₂ crystal structure with the (NIAU)₂1 one. It is not clear why there is no comparison with the (NIAU)₂2 one the authors determined using cryo-EM. Furthermore, there is a clear lack of thorough structural analysis when reporting the 6.4° rotation of the ISCU2 position when compared to the ISCU1 position. Indeed, in addition to missing to mention whether such rotation was already observed in the cryo-EM structure, nothing is described in the manuscript to take into account variations that could be induced by differences in crystal packing.

The predominant comparison of the two crystal structures was mainly due to the fact that in this manuscript we aimed to find clues for the function of the N-terminal Tyr35, and therefore the comparison to the ISCU1-containing structure (ISCU1 lacks Tyr35) was particularly interesting to see the potential influence of this (and other N-terminal) residues on the overall structure. Since we found no major differences in the overall structures, we interpret this finding as a clear sign that Tyr35 plays its essential role rather independently and not by long-range structural impacts. This is clearly supported by a newly solved crystal structure of (NIAU)₂ with ISCU2-Y35D (PDB ID: 7RTK; updated Supplementary Table 2). We now briefly discuss this issue in Results. In further response to the Reviewer's suggestion, we mention a comparison to the cryo-EM structure (Supplementary Fig. 4). Overall, we tend to believe that the differences in orientation of ISCU1/2 relative to NFS1 may reflect different environmental and/or crystallization conditions. The text has been appropriately modified.

4- The main conclusion from the crystallography analysis seems that the Y35 residue is disordered in the crystal and may adopt two favored conformations. This is not really clear from the electron density map presented in Fig 2c. Stereo-views might help better figure out the validity of the proposed model. However, there are apparently few blobs and no details

are provided by the author to support their current model. About Fig 2c, the comparison of the three variants is even more difficult to assess because the panels do not correspond to the same orientation. In addition, it is really often that N- and/or C-terminal stretches are disordered (as already observed for ISCU1). Therefore, it is really difficult to conclude anything from the presented structural data.

We apologize that apparently our 2D rendition of the two locations of Tyr35 in the various structures was not optimal. This issue will be much clearer, if the relevant pdb files are inspected. As suggested by the Reviewer, we have redone Fig. 2c, and we have added a stereo version in Supplementary Fig. 5c. We fully agree with the Reviewer's argument that, in general, terminal residues are frequently disordered and do not necessarily fulfil an important function. In case of Tyr35, our functional data suggest that this flexibility enables this residue to make contact with Tyr35 of another ISCU2 and thereby trigger dimerization. Thus, the obtained structural data makes full sense in terms of function.

5- The rationale supporting the different variants YHKK versus LSTQ or LHTQ is not really clear. What the authors really expected from these variants? In the present version of the manuscript, the reviewer feels that the authors performed different variants to mimic the ISCU1 protein (despite the orientation of the ISCU protein toward the NSF1 core was different) and that in the end they try to fit the corresponding structure into another story, yet not really convincingly. Indeed, as stated by the authors, the only relevant conclusion from this structural part is: "the high resolution 3D architecture of the (NIAU2)₂ complexes shows a flexible N-terminal region, yet otherwise is highly similar to the (NIAU1)₂ structure." The rest of the discussion sounds highly speculative and poorly supported by any data. This feeling is further supported by the M140I variant that has nothing to do with the present study.

The sequences of ISCU1 and ISCU2 differ in the first four residues of the ISCU2 sequence. With the mutations, we wanted to see if these sidechain differences have any effect on the mobility of Tyr35. From the various structures, we learned that the ISCU2 sequence contributes to the flexibility of the N-terminus. In particular, the presence of the His36 sidechain adds flexibility, and thereby helps Tyr35 to fulfil its dimerization function. To meet the Reviewer's concern, this issue is better explained, and the figures (Fig. 2c and Supplementary Fig. 5c) were redone also in stereo mode and structures are shown in the same orientation. The M140I mutant is mentioned in the text for two reasons: (1) this structure shows that this mutation, which in yeast neutralizes the need for Yfh1 (yeast frataxin homologue), introduces no structural change to the complex and (2) this crystal diffracted to the highest resolution of 1.57 Å, and as such warrants the PDB deposition and a mention in the manuscript.

6- Regarding the role of W97, the reviewer disagrees with the authors' conclusions. Extended Fig 6b indicates that the activity of the W97K variant is not affected, while that of the W97D is. The authors seem to neglect the possibility that lysine may establish a cation-pi interaction, while aspartate cannot. At least a crystal structure of the W->K variant would be expected to further investigate that possibility and see whether this stabilizes Y35 in one orientation.

Theoretically, this is a very valid point. However, the presented data shows that the possible stacking interaction between Tyr35 and Trp97 is marginal contributing at best a factor of 2 to the affinity of the complex (Fig. 3a). Moreover, we showed in Fig. 4a that the affinity of ISCU2-Y35A for (NIA)₂ complex is wild-type excluding a cation-pi interaction. We also would like to note that in the bacterial IscS-IscU complexes the equivalent tyrosine points away from the residue equivalent to Trp97 and makes contact to a Met within IscU itself. Moreover, the corresponding NFS1-W97A yeast mutant (strain Nfs1-W138A) shows normal

growth, clearly refuting a decisive role of W97 (Fig. 3e). We further provide new data using the NFS1-W97A mutant also for biochemical analysis by the in vitro enzymatic reconstitution assay (Supplementary Fig. 7c). The almost wild-type activity of NFS1-W97A again makes an impact of a cation- π interaction unlikely. Overall, it appears to us that the negatively charged Asp in the W97D variant may exhibit a harmful effect in this functional assay (Fig. 3d). The relevant text was improved.

7- In the discussion section, the authors claim that: “Our new findings together with numerous published studies (reviewed in 4) allow us to propose a molecular mechanism for the complete cycle of mitochondrial [2Fe-2S] cluster biogenesis”. In fact, the current contribution is limited to the identification of the role of Y35 as a key residue to stabilize ISCU2 dimerization. All the other parts of the proposed mechanism were already known. Agreed, but this is why we wrote “...together with numerous published studies (reviewed in 4)...”. However, we do not think it is a minor contribution to clarify how [2Fe-2S] cluster formation mechanistically occurs, and we pinpoint this reaction down to the essential function of a single conserved Tyr. We note that our manuscript also reports the first high resolution structures of the (NIAU2)₂ complex and provides cell biological evidence for various ISCU2 mutants.

Strikingly, the authors never mention, in the text, when and how iron comes in ISCU. They only mention it in Fig 6b. There isn't any mention of a possible role of iron binding and persulfide transfer and reduction in the signaling for triggering dimerization. Did the authors test for spontaneous ISCU2 dimerization in the absence of Lcysteine and/or iron?

We actually have mentioned several times in our manuscript that ISCU2 is entering the biosynthesis reaction as an iron-loaded protein (following D'Autreaux' 2019 work). In the Abstract we write that “Initial iron and sulfur loading onto monomeric ISCU2 have been elucidated biochemically, ...”, and later cite the relevant papers. In Introduction, we state “... persulfide transfer from NFS1 to one of the three conserved Cys residues of iron-containing ISCU2 bound at opposite tips of the NFS1 dimer...” (now changed to “... iron-containing ISCU2 binding to opposite tips of the NFS1 dimer” to make this clearer). In Discussion, we mention “... it was unknown how the ISCU2-bound iron and persulfide are converted into the [2Fe-2S] cluster...” and a bit later: “Persulfide transfer from NFS1 to Cys138 of iron-loaded ISCU2 is stimulated by binding of FXN (step 2)^{31,36}.”). Admittedly, we have not discussed the alternative view that FXN might bring in the iron (as this had been suggested before D'Autreaux' recent findings). This aspect was discussed intensely in our recent review (*Annu. Rev. Biochem.* 2020) and the reader is referred to this paper.

Experimentally, we already had addressed the Reviewer's question (last sentence) in our manuscript: ISCU2 in its apoform does NOT dimerize (unless artificial disulfide bonds are formed). Please see our previous Suppl. Fig. S2e (now Supplementary Fig. 12e) and Fig. 5d (which may have been overlooked by the Reviewer).

Reviewer #2 (Remarks to the Author):

Freibert et al. describe the characterization of a variant of IscU that is defective in participating in the initial steps of Fe-S cluster assembly in the mitochondria. This contribution from the Lill's group builds on their previous work and advances our understanding of how Fe-S clusters are formed in eukaryotic systems. Overall, the findings described in this study are in line with earlier reports in the literature that identified the conserved tyrosine residue at the N-terminal of IscU and reported the formation of a 2Fe-2S

cluster on an IscU dimer. The experimental data and presentation are well-sounded to support this major finding but do not explain all the aspects of the detailed model put forward by the authors. The manuscript narrative, however, makes some additional general statements that go beyond the scope of this study and, in the end, weakens the overall quality of this report. While merits publication, this contribution is built on previous findings reported and proposed by others and attempts to sell a model for a process that has not been completely elucidated. My overall recommendation is to revise the document to address these concerns and provide a discussion narrative that is more conservative and focused on the results provided in this study.

We thank the Reviewer for the overall positive evaluation and the thoughtful criticism which helped us to improve our manuscript. As mentioned above (Reviewer 1, point 1), pioneering work from the labs of Dean and Johnson showed that a stable [2Fe-2S] cluster can be bound by bacterial dimeric IscU (based on Fe and S stoichiometries and protein-FeS extinction ratios; now cited). It was not known, however, that and how cluster binding ‘induces’ (triggers) dimerization because gel filtration experiments were not performed in these early studies. We note (see above) that mitochondrial ISCU2 dimer formation was suggested by us based on Fe and S stoichiometries, FeS cluster saturation, and crosslinking studies (see Webert et al., 2014; supporting earlier work on bacterial ISC proteins). Generally, the Reviewer is concerned about the generalization of our findings on mitochondrial ISCU2 onto the bacterial system as discussed in one paragraph of Discussion. We admit that, at this point and without similar reconstitution experiments for the bacterial core ISC system, our statements have a degree of speculation. We therefore have deleted the respective paragraph. We also note that the ongoing speculation, that bacterial and mitochondrial ISC systems may be different, makes it necessary to examine both the full cell biology (*in vivo*) and biochemistry (*in vitro*) of bacterial (and archaeal) IscU as done here. We truly hope that with our new experimental data and better explanations, the Reviewer will be able to support publication of our work.

Main points

1- While the biochemical data provided in this study is very compelling, the authors have not explored how this variant may affect the involvement of frataxin and ferredoxin in earlier stages of this process. Experiments described in Fig.3AB and 4AB were completed in the absence of iron, which is known to affect the kinetic profile of these reactions (ref 32, 36). Another aspect that was not explored was the potential involvement of this residue in the reduction step promoted by ferredoxin. Thus, structural and mechanistic information is limited to investigate sulfur transfer and cluster accumulation and not the synthesis of the cluster itself. No experimental data is provided to show that IscU2 monomer, while in a complex, interacts with another monomer of IscU from a secondary core complex (Fig.6 step 3).

We assume that by “this variant” the Reviewer means ISCU2-Y35A. Concerning the suggestion by the Reviewer to explore the behavior of ISCU2-Y35A in steps requiring FXN and FDX2 function, we note that we already have explored the stimulation of (NIA)₂ complex-mediated sulfide production by FXN (and ISCU2) and the sulfur transfer from NFS1 to ISCU2 (see Fig. 4b,c). In both cases, we found no difference for the Y35A variant relative to wild-type. Concerning the involvement of Y35 in the reduction step promoted by ferredoxin, we have performed this experiment with ISCU2-Y35A as suggested, and find no major changes compared to wild-type ISCU2. This important result is now shown as new Fig. 5a.

Concerning the interaction of two ISCU2 within two different (NIAU2)₂ complexes, we have added new data to directly show that only ISCU2 proteins prebound to the (NIA)₂ complex (and not free apo-ISCU2) can give rise to the holo-ISCU2 dimer (Fig. 6b,c and

Supplementary Fig. 10; see also explanations for Reviewer 1, point 1). This data unequivocally demonstrates that only the (NIA)₂-bound form of ISCU2 and not free ISCU2 can give rise to product formation. Moreover, the new experiment also shows that apo-ISCU2 added to the (NIAU)₂ complex (in the presence of FXN and FDX2) can bind to (NIA)₂ once the bound (His-tagged) version of ISCU2 has left the complex for dimerization and [2Fe-2S] cluster formation.

2- While dimerization of IscU monomers, each coordinating 1S and 1Fe has been proposed earlier (ref 36), the data shown in this study provides a good contribution as it defines the role of Y35 as a structure element enabling this interaction. However, some of the experimental design (when indicated) to probe these reactions are performed under sub-stoichiometric amounts of IscU2 (40 uM IscU2 vs 80 uM of the NIA). Since free IscU2 is a product of this biosynthetic cycle (Fig6 step 2), one is left with a question about the effect of this reaction if an excess of IscU2 increases the rate of this reaction. How can the authors rule out a model that involves a free form of IscU (apo or Fe-bound) from interacting with the core complex and leading to the formation of an IscU dimer?

We have already published an ISCU2 titration for formation of a bridging [2Fe-2S] cluster on the ISCU2 dimer in 2014 (Webert et al. Nat Commun.). Increasing the ISCU2 concentration linearly increases the amount (not the rate) of Fe/S cluster formation. The rate limiting factor was shown earlier to be NFS1.

As to the question of "... a free form of IscU (apo or Fe-bound) from interacting with the core complex...", we have explained this above (see Fig. 6b,c and Supplementary Fig. 10). We do not find any free apo-ISCU2 as being part of the final product, i.e. holo-ISCU2.

3- The authors have not provided a comparative study to justify general comparisons with various bacterial systems that likely employ distinct mechanistic strategies to assemble clusters. While attempting to generalize this process, the authors create a problem for themselves since additional literature available for other systems not necessarily fits their proposed model. Therefore, I strongly recommend the authors to limit the discussion to the system explored in this study and minimize speculation on whether these findings are applicable to other systems.

As suggested, we have deleted the relevant paragraph in Discussion to avoid proposing that the mitochondrial and bacterial mechanisms of [2Fe-2S] cluster biogenesis are identical. This speculative discussion may be better suited for a review which would allow a much broader discussion of the pros and cons for this model, and to outline possible research directions to better resolve this question.

3.1 As noted, Tyr35 is conserved in other species containing IscU and IscU like sequences. As previously reported (ref 40), this equivalent residue in bacteria also affects function. However, suppressor mutants were identified containing substitutions within IscS A349. This evidence points to the potentially distinctive role for this residue in bacterial IscU and perhaps mechanistic differences in prokaryotic systems.

This is indeed an interesting genetic observation, but likely does not reflect a direct biochemical interaction, because A349 (of IscS) is quite far away from Y3 (IscU) in the bacterial core ISC complex (see Figure below). As Takahashi's group found, the affinity between IscS and IscU-Y3L is weakened 10-fold (rel. to IscU-wild-type), and this is restored to wild-type affinity by the IscS-A349V mutant. We therefore suspect long-range interactions within the bacterial IscS-IscU complex that affect the IscS-IscU interaction. Notably, we do not see any weakening of the (NIA)₂ affinity to all tested ISCU2-Y35 mutants (rel. to ISCU2-wt; Fig. 4a) suggesting that similar long-range effects are not relevant in the mitochondrial ISC system. This difference may fit well to the Reviewer's view of a potential difference of

the two (mito/bact) ISC systems. At any rate, deciphering the molecular basis of the A349-Y7 genetic interaction is Dr. Takahashi's project, and we will not interfere with his work.

Figure, for reviewers only:

***E. Coli* IscS-IscU complex (pdb: 3LVL)**

3.2 Characterization of prokaryotic IscU proteins has demonstrated the ability of monomeric forms of these proteins to coordinate Fe-S clusters (from *in vivo* and *in vitro* forms). These proteins can then form a 4Fe-4S cluster in reactions involving ferredoxin. Relevant to the Y35 role, recent work by Takahashi and co-workers reported on the structure of a dimer of IscU coordinating 2 [2Fe2S] clusters (<https://doi.org/10.1021/acs.biochem.1c00112>). Collectively these reports suggest that bacterial ISC systems, which do not have additional Fe-S cluster biosynthetic components like the mitochondria ISC, can coordinate a 2Fe-2S per monomer and promote the synthesis of a 4Fe-4S cluster.

This interesting new work from Dr. Takahashi's group was published during the reviewing process, and presents a holo-IscU structure (PDB 7C8M) with bound [2Fe-2S] cluster. We note, however, that the Euryarchaeon *Methanotrix* that Dr. Takahashi used for isolating his "IscU" possesses a SufSBC system rather than an ISC system. SUF systems are able to generate [4Fe-4S] clusters itself, thereby bypassing the need for a classical IscU. The physiological role of this "IscU" protein therefore is unclear and needs detailed *in vivo* investigation (a SufU-like function is not excluded, despite closer similarity to IscU). Interestingly, in none of the neighboring monomers in the unit cell of this new structure, the N-terminal Tyr are in close vicinity. Therefore, the efficient crosslinking of these Tyr, as observed in our work (Fig. 6a), is not reflected by this structure. In a subset of neighboring monomers (note that there are other interacting monomer pairs with a much larger interaction surface) two [2Fe-2S] cluster are very close to each other, and can be fused by dithionite to a [4Fe-4S] cluster. Yet, Takahashi's paper did not provide any functional data that this fusion would occur under physiologically relevant conditions, e.g., with the help of ferredoxin, or be of importance for *in vivo* [4Fe-4S] protein generation. Fdx-mediated [2Fe-2S] cluster fusion has been shown earlier by Chandramouli and Johnson in 2007, yet the fusion reaction with Fdx was much less efficient than that with dithionite, and the reaction contained DTT (which is known to reductively fuse clusters). Nevertheless, based on these *in vitro* data it is possible that in the bacterial ISC system both [2Fe-2S] and [4Fe-4S] clusters are generated on IscU. In mitochondria, however, we have shown that *in vivo* all tested mitochondrial [4Fe-4S] proteins

strictly depend on ISCA1/2 and IBA57 proteins for their maturation (Mühlenhoff JBC 2011 for yeast and Sheftel MBoC 2012 for human). Recently, we have shown *in vitro* that [2Fe-2S] cluster fusion to a [4Fe-4S] cluster requires IBA57-dependent, reductive fusion of two [2Fe-2S] cluster on the ISCA1/2 proteins by FDX2 (Weiler 2020 PNAS). These *in vivo* and *in vitro* data strongly suggest a model that [4Fe-4S] cluster formation in mitochondria does not efficiently occur *in vivo*. We now have cited and discussed the new structure. We also mention that this structure would not readily provide a clue why the N-terminal Tyr would be such a functionally important residue for interaction-induced dimerization.

3.3 The discussion about the potential role of SufU in functioning as a scaffold disregards recent literature describing genetic, biochemical, and structural differences between bacterial IscU and SufU from a number of species. The location of equivalent Y in the SufS-SufU structure is likewise remote from the site of sulfur transfer. While it may affect interactions with SufS, it remains to be determined if substitution of this residue will affect zinc binding and sulfur transfer reactions to downstream scaffold SufBCD, which is absent for the species investigated in this study.

We certainly do not question the current literature on SufU as a Zn-dependent sulfur transferase, just raise the question why the N-terminal Y is conserved in SufU. We admit that this topic does not directly relate to our current story, and is speculative. Therefore, we decided to delete this paragraph in Discussion.

3.4 The structure of the archaea A. fulgidus IscS-IscU complex shows the coordination of an Fe-S cluster to a monomer of IscU in which a cysteine residue of IscS provides the fourth ligand. The authors are correct to indicate the IscS is inactive as it lacks the conserved lysine coordinating the PLP(PMP) cofactor. In this particular case, it has been proposed that the activity of a cysteine desulfurase may be dispensable due to the organism's lifestyle. Regardless, this structural model provides a valid example that an IscU monomer is capable of coordinating a cluster. And again, this structure highlights mechanistic differences from eukaryotic and prokaryotic systems and perhaps suggests their functions have evolved to meet distinct biosynthetic requirements and environmental factors.

We do not deny the possibility that the archaeal-bacterial ISC systems may mechanistically differ from the mitochondrial one. However, until an *in vivo* study has been performed to actually verify the suggested mechanism, a definitive conclusion cannot be drawn. This concern does not affect our proposed mitochondrial mechanism which we think is supported from multiple angles in our manuscript. We further would like to emphasize that we never found a Fe/S cluster directly bound to the core ISC complex. Furthermore, the available structures of IscU with bound [2Fe-2S] cluster show (so far unexplained) differences in the sidechains liganding the cluster (His vs. Asp) and a resulting different orientation of the clusters in these structures. We have modified relevant text to remove unclear statements.

Overall, a suggested approach for revising the discussion is to acknowledge that these systems likely evolve to perform distinct functions. The case of Mycoplasma noted in the manuscript is a good example of that. Prokaryotic Fe-S assembly is not confined to organelles and employs considerably fewer components. Therefore these components may accumulate multiple and distinct functions. Presenting a narrative that invalidates or discredits the findings reported for other systems without providing experimental evidence does not improve the impact of this publication or inspire others to define unresolved aspects of these complex pathways.

We agree with this point and, as noted above, have removed our speculative statements concerning the similarities of the mechanisms of bacterial and mitochondrial [2Fe-2S] biosynthesis. Nevertheless, we would like to note that the core ISC complex is fairly well

conserved between mitochondria and bacteria. In fact, the core ISC components differ only in ISD11-ACP, a regulatory sub-complex only present in mitochondria, and bacterial IscX not present in mitochondria. Moreover, Tyr35 is universally conserved. At any rate, this interesting issue raised by the Reviewer can only be solved by future dedicated comparative experiments.

Minor points

- *Figure 3C legend or figure needs to indicate that frataxin was added.*

This information was already provided in Materials and Methods. As requested, we now have added this information also to the first legend presenting the persulfide transfer.

- *Methods describing cluster enzymatic reactions do not include a description of reaction ingredients and concentrations. Please revise.*

This information was added to Materials and Methods.

- *Page 10 second paragraph discussion – new high resolution structures (as low as 1.57 Å)*
Amended.

REVIEWERS' COMMENTS

Reviewer #1 (Remarks to the Author):

The reviewer has carefully read the revised version of the manuscript from Dr Svein-A Freibert and co-workers. All of the previously raised concerns have been fully addressed in the significantly improved new version. Therefore, the reviewer does not have any further concerns and fully support this manuscript for publication in Nature Communications. The reviewer would like to congratulate the authors for their work.

Reviewer #2 (Remarks to the Author):

In this revised version, Freibert et al. provide a much-improved version of the manuscript reporting the involvement of the Y35 residue during the formation of Fe-S on ISCU2. I appreciate the author's best intentions to address the concerns raised in the first round of reviews. However, minor points still need to be clarified before acceptance of this manuscript for publication.

1) On page 9, line 18 and discussion, first paragraph– The authors describe the reductive cleavage of ISCU2 persulfide by FDX2. This result conflicts with Gervason et al. (in Nat Comm 2019) who showed that Fdx-dependent reduction of persulfide on IscU was dependent on the presence of Fe. Since Fdx promotes one-electron transfer reactions and persulfide reduction is a two-electron reduction, how do the authors reconcile persulfide reduction in the absence of Fe? Worth noting that in this same study (Gervason, 2019), the authors characterized the final product as [2Fe-2S]-IscU monomer. My suggestion is to elaborate on the presented finding in the context of other proposals.

2) On page 10, last paragraph – The experimental design to probe the source of the second monomer of IscU2 during dimer-cluster formation does not rule other possibilities. First, an excess of untagged free ISCU2 was used. Since the experimental design only probed the his-tagged version, one cannot determine if the untagged IscU has also engaged in the formation of the holo dimer. Perhaps if the reverse experiment was completed (e.g. free hisISCU2), additional support for this proposal would have been provided. Also, the experimental design does not determine if two ISCU2 from the same complex or different complexes interact during dimer formation. While the structures determined by the authors before and in this study show the two monomers of ISCU2 in a conformation that does not allow such direct interaction, other structures (Cory et al.) have suggested that alternate arrangements may be possible. Although not preferred by this reviewer and the author, it is possible that NAI complex undergoes a conformational change during cluster

assembly leading to interaction between ISCU2 monomers within the complex. Thus, crystal structurals provide a snapshot of a dynamic process with several moving parts.

3) Page 13, end of the page – While the authors reference a recent structure of holo-IscU dimer containing two 2Fe-2S cluster (ref 51), the authors do provide a somewhat misleading description of the involvement of conserved tyrosine (Y7) in dimer formation. It is accurate to indicate that the two Tyr residues from each monomer are pointing to opposite directions, but it is important to indicate that this residue is engaged in a pi stacking with a conserved Phe (F61) from the other subunit. Interestingly Phe or Tyr are found at this conserved position suggesting that this specific Pi stacking interaction during dimer formation could be conserved in other IscU sequences.

Response to the requests of Reviewer #2

(Remarks to the Author):

In this revised version, Freibert et al. provide a much-improved version of the manuscript reporting the involvement of the Y35 residue during the formation of Fe-S on ISCU2. I appreciate the author's best intentions to address the concerns raised in the first round of reviews. However, minor points still need to be clarified before acceptance of this manuscript for publication.

We thank the Reviewer for acknowledging the improvement of our manuscript by adding new data and providing better explanations. The new minor points are addressed below.

1) On page 9, line 18 and discussion, first paragraph– The authors describe the reductive cleavage of ISCU2 persulfide by FDX2. This result conflicts with Gervason et al. (in Nat Comm 2019) who showed that Fdx-dependent reduction of persulfide on IscU was dependent on the presence of Fe. Since Fdx promotes one-electron transfer reactions and persulfide reduction is a two-electron reduction, how do the authors reconcile persulfide reduction in the absence of Fe? Worth noting that in this same study (Gervason, 2019), the authors characterized the final product as [2Fe-2S]-IscU monomer. My suggestion is to elaborate on the presented finding in the context of other proposals.

As originally noted by Gervason et al. persulfide transfer only works efficiently in the presence of iron. The Reviewer may have missed that our persulfide transfer assays contain 200 μM FeCl_2 (see Materials and Methods). In consequence, there is NO conflict between our findings and those of Gervason et al.

2) On page 10, last paragraph – The experimental design to probe the source of the second monomer of IscU2 during dimer-cluster formation does not rule other possibilities. First, an excess of untagged free ISCU2 was used. Since the experimental design only probed the his-tagged version, one cannot determine if the untagged IscU has also engaged in the formation of the holo dimer. Perhaps if the reverse experiment was completed (e.g. free hisISCU2), additional support for this proposal would have been provided. Also, the experimental design does not determine if two ISCU2 from the same complex or different complexes interact during dimer formation. While the structures determined by the authors before and in this study show the two monomers of ISCU2 in a conformation that does not allow such direct interaction, other structures (Cory et al.) have suggested that alternate arrangements may be possible. Although not preferred by this reviewer and the author, it is possible that NAI complex undergoes a conformational change during cluster assembly leading to interaction between ISCU2 monomers within the complex. Thus, crystal structural snapshots provide a snapshot of a dynamic process with several moving parts.

The Coomassie-stained blue gel (Fig. 6c, right) shows that there is virtually no non-tagged ISCU2 in the dimer fraction (i.e. holo-protein), ruling out that untagged ISCU2 was matured to a holo-protein. On the other hand, the anti-HIS Western blot analysis (Fig. 6c, left) clearly shows no HIS-tagged ISCU2 in the monomeric fraction suggesting that virtually all $(\text{NIA})_2$ complex-bound HIS-ISCU2 was matured into dimeric holo-protein. While designing the optimal experimental setup to answer the Reviewer's question from the initial round, we have varied the amount of unbound non-tagged ISCU2 needed for these experiments. If we used only one (instead of two, as shown in Fig. 6b) equivalent of non-tagged ISCU2, we see virtually all non-tagged ISCU2 bound to the $(\text{NIA})_2$ complex after dissociation of His-tagged ISCU2, yet no free monomeric non-tagged ISCU2. Since the monomer, however, is nice to present as a marker peak, we decided to show the results for this stoichiometry, because it contains more information. The overall result, however, stays the same: Only $(\text{NIA})_2$ -bound HIS-tagged ISCU2 and not the free non-tagged ISCU2 form the dimer, while the previously unbound non-tagged ISCU2 can now bind to the $(\text{NIA})_2$ complex and excess non-tagged ISCU2 stays as a monomer. We therefore do not see how the result can be interpreted differently. Reversal of non-tagged and HIS-tagged

ISCU2 in principle is possible, but we know that both proteins are fully active, and such an experiment would only make sense, if one doubts that the HIS-tagged protein cannot undergo the dimerization.

Concerning the speculation of large structural changes of the (NIA)₂ complex, we doubt – as the Reviewer – that such internal (NIA)₂ complex transitions play a decisive role in vivo. This is supported by a recent presentation of Dr. Dave Barondeau (who published this ‘open’ structure in PNAS) in the Online Symposium on ‘Iron-sulfur protein biogenesis’ organized by us and Dr. F. Barras (see https://www.uni-marburg.de/en/fb20/departments/cyto/copy_of_bilder/bilder-lill/fes-test). Dave showed new data suggesting that the ‘open’ conformation of (NIA)₂ may exist in vitro, but to be active the (NIA)₂ complex has to be in the closed, i.e. standard conformation.

3) Page 13, end of the page – While the authors reference a recent structure of holo-IscU dimer containing two 2Fe-2S cluster (ref 51), the authors do provide a somewhat misleading description of the involvement of conserved tyrosine (Y7) in dimer formation. It is accurate to indicate that the two Tyr residues from each monomer are pointing to opposite directions, but it is important to indicate that this residue is engaged in a pi stacking with a conserved Phe (F61) from the other subunit. Interestingly Phe or Tyr are found at this conserved position suggesting that this specific Pi stacking interaction during dimer formation could be conserved in other IscU sequences.

We have noted this interaction when we inspected the Takahashi structure of bacterial IscU. Whether this is a biologically relevant pi stacking interaction (or a consequence of crystal packing) remains to be tested. Since we can replace Y35 (of ISCU2) by a mixture of ISCU2-Y35D and -Y35K and get full biological activity in [2Fe-2S] cluster formation, this potential interaction would have to occur during a different step of the entire process.